

# A Hybridized NGBoost-XGBoost Framework for Robust Evaporation and Evapotranspiration Prediction

Hakan Başağaoğlu[1,*], Debaditya Chakraborty[2,*], and James Winterle[1]

[1]Edwards Aquifer Authority, San Antonio, TX 78215, USA
[2]University of Texas at San Antonio, San Antonio, TX 78207, USA
[*]These authors contributed equally to this work.

**Correspondence:** Hakan Başağaoğlu (hbasagaoglu@edwardsaquifer.org)

**Abstract.** We analyze the relationship between potential evapotranspiration ($ET_o$), actual evapotranspiration ($ET_a$), and surface water evaporation ($E_{sw}$) in the semi-arid south-central Texas, using hourly climate data, daily lake evaporation measurements, and daily actual evapotranspiration measurements from an eddy covariance (EC) tower. The deterministic analysis reveals that $ET_o$ set the upper bound for $ET_a$, but the lower bound for $E_{sw}$ in the study area. Unprecedentedly, we demon-

strate that a newly developed probabilistic machine learning (ML) model, using a hybridized NGBoost-XGBoost framework, can accurately predict the daily $ET_o$, $E_{sw}$, & $ET_a$ from local climate data. The probabilistic approach exhibits great potential in overcoming data uncertainties, in which 99% of the $ET_o$, 90% of the $E_{sw}$, and 91% of the $ET_a$ test data at three watersheds were within the model's 95% prediction interval. The probabilistic ML model results suggest that the proposed framework can serve as a robust and computationally more efficient tool than the hourly Penman-Monteith equation to predict the $ET_o$ while

avoiding computationally-involved net solar radiation calculations. Additionally, the performance analysis of the probabilistic ML model indicates that it can be successfully implemented in practice to overcome the uncertainties associated with pan evaporation & pan coefficients in $E_{sw}$ estimates, and to offset the high capital & operational costs of EC towers used for $E_a$ measurements. Finally, we demonstrate, for the first time, a coalition game theory approach to identify the order of importance, dependencies & interactions of climatic variables on the ML-based $ET_o$, $E_{sw}$, and $ET_a$ predictions. New knowledge

gained through the game theory approach is beneficial to strategically locate weather stations for enhanced evapo(transpi)ration predictions, and plan out sustainability and resilience efforts, as part of water management and habitat conservation plans.

## 1 Introduction

Evapo(transpi)ration is one of the key components of a groundwater budget in drought-prone regions with scarce water supplies (Heilman et al., 2009; Gokmen et al., 2013; Glenn et al., 2015), facing challenges of sustainable development and climate

resilience. Reliable prediction of evapo(transpi)ration is useful in such regions to determine aquifer recharge (Hauwert and Sharp, 2014; Xie et al., 2018), and subsequently, evaluate groundwater sustainability to meet municipal, agricultural, ranching, industrial, and recreational water demands, while sustaining quality & quantity of environmental flows to protect and maintain a healthy ecologic environment for endemic groundwater-obligated species.





The karstic Edwards aquifer in semi-arid south-central Texas is the primary source of drinking water for the city of San Antonio and is also home to several threatened and endangered aquatic species (e.g., Texas blind salamander, San Marcos salamander) at the major spring outlets (Devitt et al., 2019). Up to 65% of rainfall is lost to evapo(transpi)ration (Dugas et al., 1998) in south-central Texas, which has a few permanent surface waters and experiences frequent droughts. In some years, anomalously sinking motions and divergent water vapor flux over the Texas area reduce precipitation and increase downward solar radiation, which results in dry and hot soil promoting the occurrence of extreme heat waves (Deng et al., 2018). Such an extreme summer heat wave occurred in 2011 with average temperature 3°C above the 1981-2010 mean for June through August (Hoerling et al., 2013). The likelihood of exceeding a given unusually high summer temperature in the Texas region was reported to be about 10 times greater with 2011 anthropogenic emissions compared to preindustrial forcing (Rupp et al., 2015). Under the current and forecasted climate conditions in south-central Texas, increased air and groundwater temperatures and decreased aquifer recharge and springs flow could make endangered or threatened endemic aquatic species vulnerable to extinction (Mahler and Bourgeais, 2013; Devitt et al., 2019). Therefore, reliable estimates of evapo(transpi)ration are essential for improved management of Edwards aquifer's groundwater resources and environmentally-sensitive habitats for groundwater-dependent species, as part of the current and future resource planning. In light of the importance of evapo(transpi)ration processes for the karstic aquifer system in south-central Texas, the main objectives of this paper are to

1. investigate the relation between the potential evapotranspiration ($ET_o$), surface water evaporation ($E_{sw}$), and the actual evapotranspiration ($ET_a$) in the semi-arid region of south-central Texas,

2. develop novel ML-based probabilistic predictive models of $ET_o$, $E_{sw}$, $ET_a$ based on local climate data, and assess the models' predictive performances using statistical measures, and

3. apply a game theory approach to determine the order of importance, dependencies & interactions of climatic variables on $ET_o$, $E_{sw}$, and $ET_a$ predictive models.

Recently, several ML models (e.g. genetic algorithms, neural networks, clustering, tree-based ensembles, fuzzy models, multivariate adaptive regression splines, extreme learning machines) have shown promising results due to their ability to simulate the complex nonlinear behavior of the reference evapotranspiration, pan evaporation, terrestrial evapotranspiration (Nema et al., 2017; Feng et al., 2017; Lu et al., 2018; Jovic et al., 2018; Dou and Yang, 2018; Mehdizadeh, 2018; Kisi and Alizamir, 2018; Tao et al., 2018; Fan et al., 2018; Sanikhani et al., 2019; Pan et al., 2020). However, a critical challenge with these existing ML models is that the nonlinear relationship between climatic variables and the evapo(transpi)ration makes it difficult to account for inherent uncertainties (Tang et al., 2018). Therefore, in this paper, we confront the uncertainties in evapo(transpi)ration predictions using a hybrid probabilistic NGBoost-XGBoost ML model without compromising the accuracy of the predictions. The probabilistic model takes in respective feature values $x$ and returns a distribution over the target $y$ indicating the relative likelihood of different values of $y$. To our knowledge, ML-aided *probabilistic predictions* of $ET_o$, $E_{sw}$, and $ET_a$ is unprecedented. We demonstrated that the hybrid ML model is capable of producing robust and accurate daily $ET_o$, $E_{sw}$, and $ET_a$ predictions based on historical climate data, in which $\geq 90\%$ of the predicted target values were within the 95% prediction interval.





Moreover, for the first time we applied a game theory approach (Lundberg et al., 2020) to explain the importance of the features (e.g., climatic variables) on the ML-based $ET_o$, $E_{sw}$, and $ET_a$ predictions. This approach manifests how the individual

feature value, while considering its interaction with other features learned and built-up from the historical data, influences the model's predictions, which enhances the model's ability to make sentient projections honoring the underlying hydrological processes. Our analysis revealed that the top three most important variables in the order of importance in south-central Texas for $ET_o$ predictions are the shortwave solar radiation, air temperature, and relative humidity; for $E_{sw}$ predictions are the surface water temperature, month of the year, and relative humidity; and for $ET_a$ predictions are the shortwave solar radiation, month

of the year, and relative humidity. Such information would be useful to strategically locate weather stations and sensors over the aquifer region to collect the most relevant data for enhanced evapotranspiration predictions and/or assess the suitability of simplified evapotranspiration prediction models for the watersheds with scarce data. Moreover, the interpretability of our ML model in combination with the game theory approach is capable of revealing new knowledge that may not be immediately apparent. For example, although soil moisture content was not included in our ML-model as a feature, but its effect was

captured in $ET_a$ measurements at the EC tower, the ML model predicted low $ET_a$ despite high $ET_o$ & low $RH$ in certain times, which could be an indication of critical water deficiency in the soil. Such new knowledge from the ML model is essential for the current and future "well-informed" groundwater management and habitat conservation plans.

## 2  Methods

**Description of Evapo(transpi)ration Measures**.  Different evapo(transpi)ration measures, including pan and lake evapora-

tion, potential evapotranspiration, and actual evapotranspiration considered in this paper are briefly described here prior to associated calculations and ML methods are introduced. For more comprehensive discussion, the reader may refer to the paper by McMahon et al. (2013).

Evaporation pans are used to determine evaporation from water surface at the pan-scale ($E_p$), which are then scaled-up to estimate evaporation from open water bodies ($E_{sw}$) such as lakes (Dingman, 1992). Therefore, lake evaporation can be

interpreted as hybrid measured-estimated evaporation. $E_{sw}$ was viewed to represent regional potential evaporation (Vercauteren et al., 2009) and has been used in terrestrial water balance calculations (Roderick et al., 2009). Daily or monthly empirical Meyer's formulas (MF) have been used to calculate $E_{sw}$, based on surface water temperature, relative humidity, and wind speed measurements (Penman, 1948; Xu and Sing, 2002; Burn and Hesch, 2006).

Potential evapotranspiration ($ET_o$), on the other hand, accounts for climate-driven watershed-scale evapotranspiration from a

hypothetical reference crop in a saturated soil, which reflects the evaporation power of the atmosphere. The Penman-Monteith equation (PME), based on the energy-balance, is used to calculate $ET_o$ (Allen et al., 1998). PME calculations require time series of shortwave solar radiation, air temperature, atmospheric pressure, relative humidity, and wind speed data. PME can be used for hourly to monthly $ET_o$ estimates, depending on the temporal resolution of the input climate data. $ET_o$ was used to estimate $E_{sw}$ (Vercauteren et al., 2009), actual evaporation (Boughton, 2004), vegetation potential evapotranspiration (Jia

et al., 2009) or aridity index (Nash et al., 1997). The PME can be coupled with surface conductance models and leaf area





indices for evapotranspiration predictions. Using this approach, Zhang et al. (2008) reported good agreement between 5-year average evaporation rates predicted through the PME and using water balances at 120-gauged catchments in Australia. Leuning et al. (2008) noted reliable estimates of daily evapotranspiration rates at the kilometer-scale using the PME. PME calculations, however, are more complicated and involve more climate variables than empirical MFs. Therefore, simplified versions of the PME with fewer climate variables have been explored and tested for watersheds with scarce data (Fan et al., 2018; Irmak et al., 2003; Peng and Feng, 2017).


However, neither $ET_o$ nor $E_{sw}$ provides a direct estimate for actual evapotranspiration ($ET_a$), which is the sum of evaporation from soil and transpiration from vegetation. As compared to $ET_o$, $ET_a$ is more site-specific and spatially-variable, depending on soil and vegetation types. Reportedly, an increase in transpiration from vegetation could result in a two-fold decrease in soil evaporation (Yongqiang et al., 2016). Eddy covariance (EC) is the most direct method of measuring land surface water vapor flux (Burba, 2013) without disturbing the water-air interface (Vesala et al., 2006), and hence, provides accurate site-scale $ET_a$ measurements (Wang et al., 2015). When coupled with the energy balance method, the EC technique provides an alternative measure of latent heat flux equivalent to $ET_a$ (Wilson et al., 2001; Zitouna-Chebbi et al., 2018). Shi et al. (2008) noted that PME resulted in higher latent flux than the EC method in estimating $ET_a$ of dry forest canopy. The relation between $ET_o$ and $ET_a$ is to be explored for the semi-arid region in this paper.



**Penman-Monteith Equation (PME).** A detailed description of underlying physical processes and calculation steps of the PME for hourly $ET_o$ estimates can be found in the FAO by Allen et al. (1998). This section provides the main equations and critical implementations for the solution of the PME for hourly $ET_o$ given by

$$ET_o = \frac{0.408\triangle\left(R_n - G\right) + \gamma\frac{37}{T_a+273}u_2\left(e^o - e_a\right)}{\triangle + \gamma\left(1 + 0.34u_2\right)},\tag{1}$$

where $\triangle$ is the slope of the saturation vapor pressure [kPa $^oC^{-1}$], $R_n$ is the net solar radiation [MJ/(m$^2$ d)], $G$ is the heat flux [MJ/(m$^2$ d)], $\gamma$ is the psychrometric constant [kPa $^oC^{-1}$], $T_a$ is the air temperature [$^oC$], $e^o$ is the saturated vapor pressure [kPa], $e_a$ is the actual vapor pressure [kPa], and $u_2$ is the wind speed measured at 2 m above the ground surface [m/s]. $\gamma = 0.665 \times 10^{-3}P$, in which $P$ is the atmospheric pressure [kPa]. $R_n = (1-\alpha)R_{ns}$, in which $\alpha$ is the albedo that determines the fraction of the measured solar radiation, $R_s$ [MJ/m$^2$ d], reflected by the surface. $e^o = 0.6108e^{T_a^*}$, $e_a = e^o(RH)/100$, and

$\triangle = 2503.058e^{T_a^*}/(T_a+237.3)^2$, in which $RH$ is the relative humidity [-] and $T_a^* = 17.27T_a/(T_a+237.3)$. Hourly-averaged $T_a$, $RH$, $P$, $u_2$, $e_a$, and $e^o$, and hourly-aggregated $R_s$ are used in Eq. 1. Net solar radiation is defined as $R_n = R_{ns} - R_{nl}$, in which $R_{ns}$ is the measured net incoming shortwave radiation and $R_{nl}$ is the outgoing longwave radiation [MJ/(m$^2$ d)] computed as

$$R_{nl} = \sigma\left[\frac{T_{a,max}^4 + T_{a,min}^4}{2}\right]\left(0.34 - 0.14\sqrt{e_a}\right)\left(1.35\frac{R_s}{R_{so}} - 0.35\right),\tag{2}$$

where $\sigma$ is the Stefan-Boltzmann constant ($4.903 \times 10^{-9}$ MJ / (K$^4$ m$^2$ d), $T_{a,max}^4$ and $T_{a,min}^4$ are the maximum and minimum absolute air temperatures during the 24-hour period [K]. $R_{so}$ is the clear-sky radiation [MJ/(m$^2$ d)]. Linearized Beer's radiation





law leads to $R_{so} = \left(0.75 + 2 \times 10^{-5} z\right) R_a$, in which $z$ is the elevation of the weather station above the sea level [m] and $R_a$ is the extraterrestrial radiation [MJ/(m$^2$ d)]. In other words, $R_{so} \sim 0.75$ of $R_a$, which accounts for $25\%$ reduction in $Ra$ due to the interaction of $R_a$ with atmospheric gases (Zhang et al., 2008; Raza and Mahmood, 2018). $(R_s/R_{so})$ is the relative shortwave
radiation, representing the cloud cover, defined as

$$0.33 \leq \frac{R_s}{R_{so}} \sim \frac{R_s}{\left(0.75 + 2 \times 10^{-5} z\right) R_a} \leq 1.0, \tag{3}$$

in which the lower bound of 0.33 and the upper bound of 1.0 represent the dense cloud cover and clear sky on a particular day, respectively. The first, second, and third terms in Eq. 2 account for the effect of air temperature, air humidity, and cloudiness on $R_{nl}$. $R_a$ depends on the geographic location of the weather station and time of the day, and is computed as

$$R_a = \frac{72 G_{sc} d_r}{\pi} \left[\left(\omega_2 - \omega_1\right) \sin(\varphi) sin(\delta) + \cos(\varphi) \cos(\delta) \left(\sin(\omega_2) - \sin(\omega_1)\right)\right], \tag{4}$$

where $G_{sc}$ is the solar constant [0.0820 MJ/(m$^2 min$)], $d_r$ is inverse relative distance earth-sun [-], $\delta$ is the solar declination [rad], $\varphi$ is the latitude of the weather station [rad], $\omega_1$ and $\omega_2$ are the solar time angle at the beginning and end of the period [rad]. Here, $d_r = 1 + 0.033 \cos\left(2\pi J/365\right)$ and $\delta = 0.409 \sin\left(2\pi J/365 - 1.39\right)$, in which $J$ is the day count of the year. Solar time angle at midpoint of hourly period, $\omega$ [rad], is given by

$$\omega = \left(\pi/12\right)\left(\left[t + 0.006667\left(L_z - L_m\right) + S_c\right] - 12\right], \tag{5}$$

in which $t$ is the standard clock time at an half-and-hour intervals [hr], $L_z = 90^{\circ}$ for central Texas, $L_m$ is the longitude of the weather station [degrees], and $S_c$ is the seasonal correction for solar time [hr], given by $S_c = 0.1645 \sin\left(2b\right) - 0.1255 \cos\left(b\right) - 0.025 \sin\left(b\right)$, in which $b = 2\pi\left(J - 81\right)/364$. $\omega_1 = \omega - \left(\pi t_1/24\right)$ and $\omega_2 = \omega + \left(\pi t_1/24\right)$.

In hourly $ET_o$ calculations, $R_a = 0$ when the sun is below the horizon at $\omega < -\omega_s$ or $\omega > \omega_s$. To keep the cloudiness,
$R_s/R_{so}$ in Eq. 3, and hence, $R_{nl}$ in Eq. 2 finite, $R_s/R_{so}$ at night times (i.e., when the sun is below the horizon) is set to $R_s/R_{so}$ value 2-3 hours prior to sunset. The sunset time in each day of the year can be identified by $\left(\omega_s - 0.79\right) \leq \omega \leq \left(\omega_s - 0.52\right)$. When the sun is above the horizon $(R_a > 0)$, $G = 0.1 R_n$ corresponds to smaller heat outfluxes, promoting soil warming during day times. In contrast, when the sun is below the horizon $(R_a = 0)$, $G = 0.5 R_n$ corresponds to larger heat outfluxes, promoting soil cooling at nights. Moreover, wind speed, $u_2 \geq 0.5$ m/s in $ET_o$ calculations to account for the effects of boundary layer
instability and buoyancy of air in promoting exchange of vapour at the surface when air calm.

**Meyer's Formula (MF).** Meyer's formula (MF) is a mass transfer-based, empirically constructed formula (Meyer, 1915), whose different versions have been used to calculate daily or monthly $E_{sw}$. It is typically expressed in the form of $E_{sw} = \beta \left(e^o - e_a\right)$, in which $\beta$ is the empirically determined constant, $e^o$ and $e_a$ are defined in terms of surface water temperature,





unlike in the PME. Reportedly, the best form of the MF to predict daily $E_{sw}$ from free water surface constructed using data
from England (Penman, 1948; Xu and Sing, 2002)

$$E_{sw,d} = 0.35 \left(1 + 0.98/100 u_2\right)\left(e^o - e_a\right) \tag{6}$$

where $u_2$ is expressed in [mm/d], and $e^o$ and $e_a$ are expressed in [mm-Hg] in Eq.6. For monthly evaporation estimates from
surface of a water body using data from Canadian prairies (Burn and Hesch, 2006),

$$E_{sw,m} = 7.58 C \left(1 + 6.21 \times 10^{-2} u_{7.62}\right)\left(1 + 3.28 \times 10^{-5} z\right)\left(e^o - e_a\right), \tag{7}$$

where $u_{7.56}$ is the monthly-averaged wind speed measured at 7.56 m above the ground surface, expressed in [km/hr], and $e^o$
and $e_a$ are expressed in [mbar] in Eq.7. $C = 1$ in the original work by Burn and Hesch (2006). $C \neq 1$ is introduced here to
adjust the magnitude of monthly $E_{sw}$. When compared to PME in Eq. 1, MFs involve fewer climate variables and do not
involve computationally-involved net solar radiation calculations.

**ML Methods**.  Recently, Fan et al. (2018) showed that extreme gradient boosting (XGBoost) is capable of producing relatively
accurate predictions of daily $ET_o$ in comparison to other ML models for different climatic zones of China. However, XGBoost
provides a point prediction that does not include any information regarding the level of variability in the predicted hydrological
characteristics such as $ET_o$, $E_{sw}$, and $ET_a$. To solve this inherent problem, we propose a unique NGBoost hybridized with
XGBoost model to produce point predictions as well as a probability distribution over the entire outcome space for quantifying
the uncertainties related to hydrological predictions. The proposed hybrid model could provide practitioners with a better
understanding of the uncertainty in the $ET_o$, $E_{sw}$, and $ET_a$ predictions without compromising the accuracy of the predictions.
      XGBoost, proposed by Chen and Guestrin (2016), is a tree-based ensemble learning algorithm that follows the principle of
boosting. Boosting is a general technique in ML, where multiple weak learners such as Classification and Regression Trees
(CART) are organized to produce a strong learning model (Marsland, 2014). The fundamental concept behind this technique is
to produce new learners that are sequentially fitted to the residuals from the previous learner, which are then added to the model
to update the residuals. Gradient boosting enhances the flexibility of the boosting algorithm by generating the new learners that
are maximally correlated to the negative of the gradient of the loss function. This process enables the convergence of the loss
function and allows arbitrary differentiable loss functions to be used in the model building process (Chen, 2014; Chen and
He, 2015). From the computational standpoint, XGBoost is built with a multiprocessing OpenMP API (Chandra et al., 2001),
which enables XGBoost to use all the CPU cores in parallel during while training, making it computationally efficient and
scalable. Moreover, XGBoost presorts the independent variables at the beginning of the training process, which further reduces
the training complexity and computational time.
      NGBoost, proposed by Duan et al. (2019), is a supervised learning algorithm with generic probabilistic prediction capability.
A probabilistic prediction produces a full probability distribution over the entire outcome space; thus, enabling the users to
quantify the uncertainties of the evapotranspiration predictions produced by the model. In standard point prediction settings,





the object of interest is an estimate of the scalar function $E(y|x)$, where $x$ is the feature vector and $y$ is the prediction target, without accommodating uncertainty estimates. In contrast, under a probabilistic prediction setting, a probabilistic forecast with probability distribution $P_\theta(y|x)$ is produced by predicting the parameters $\theta$. NGBoost can perform probabilistic forecast with flexible tree-based models, given that NGBoost is designed to be scalable and modular with respect to the base learner (e.g. decision trees), probability distribution parameter (e.g. normal, Laplace), scoring rule (e.g. Maximum Likelihood Estimation).

We utilized NGBoost's modular design to hybridize it with XGBoost base learners to enhance the resulting model's predictive capability. As shown in Fig. 1, the input feature vector $x$ in the hybrid NGBoost-XGBoost model is passed on to the XGBoost base learners to produce a probability distribution of the predictions $P_\theta(y|x)$ over the entire outcome space $y$ (i.e., $ET_o$). The models are then optimized by scoring rule $S(P_\theta, y)$ using a maximum likelihood estimation function that yields calibrated uncertainty and point predictions. The feature vector $x$ for $ET_o$ and $ET_a$ predictions consists of $T_a$, $P$, $RH$, $u_2$, and $R_s$; and the feature vector $x$ for $E_{sw}$ predictions consists of $T_{sw}$, $T_a$, $P$, $RH$, $u_2$, and $R_s$.

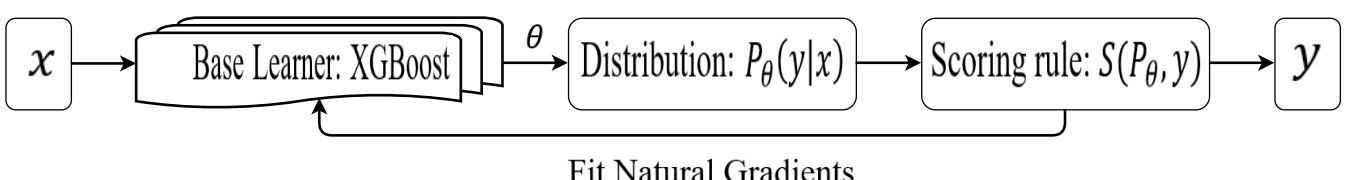

**Figure 1.** Conceptual representation of the hybrid NGBoost-XGBoost model for $ET_o$, $E_{sw}$, and $ET_a$ prediction.


## 2.1 Data Availability

The Edwards Aquifer Authority (EAA) initiated a pilot program in 2014 to establish a network of weather stations across the Edwards aquifer region to collect local climate data. Measured local climate data at these stations relevant to watershed-scale $ET_o$ calculations include the incoming shortwave solar radiation ($R_s$), atmospheric pressure ($P$), air temperature ($T_a$), relative

humidity ($RH$), and wind speed ($u_2$). For this study, local climate data at the 15 min intervals from 9/1/2015 to 12/31/2019 were acquired from weather stations at the Nueces Duernell Ranch (NDR) and Bandera County River Authority and Groundwater District's office (BCRA) in Fig. 2. Local climate data at the Camp Bullis Savanna (CBS) station was available since 1/25/2016.

**Local climate data at the NDR weather station.** For hourly-$ET_o$ calculations, hourly-averaged $T_a$, $P$, $RH$, and $u_2$ and hourly-summed $R_s$ at the NDR station, shown in Fig. 3, were used as input in Eq. 1. The total number of missing hourly

records was 2, which were filled in by linear interpolation. The NDR weather station was selected in the analysis due to its proximity to Uvalde County, TX, where monthly representative cloud cover data was available, which were used to test the model accuracy in Section 3. Local climate data at the BCRA and CBS stations, along with $ET_a$ measurements at Savanna, Well 10 are provided in Appendix A.

**Surface Water Data.** Daily and monthly surface water evaporation data closest to the NDR site were obtained from Ingram

Lake in Texas. Daily pan evaporation measurements ($E_p$) from 9/1/2015 to 12/31/2019 were taken by the Texas Water De-





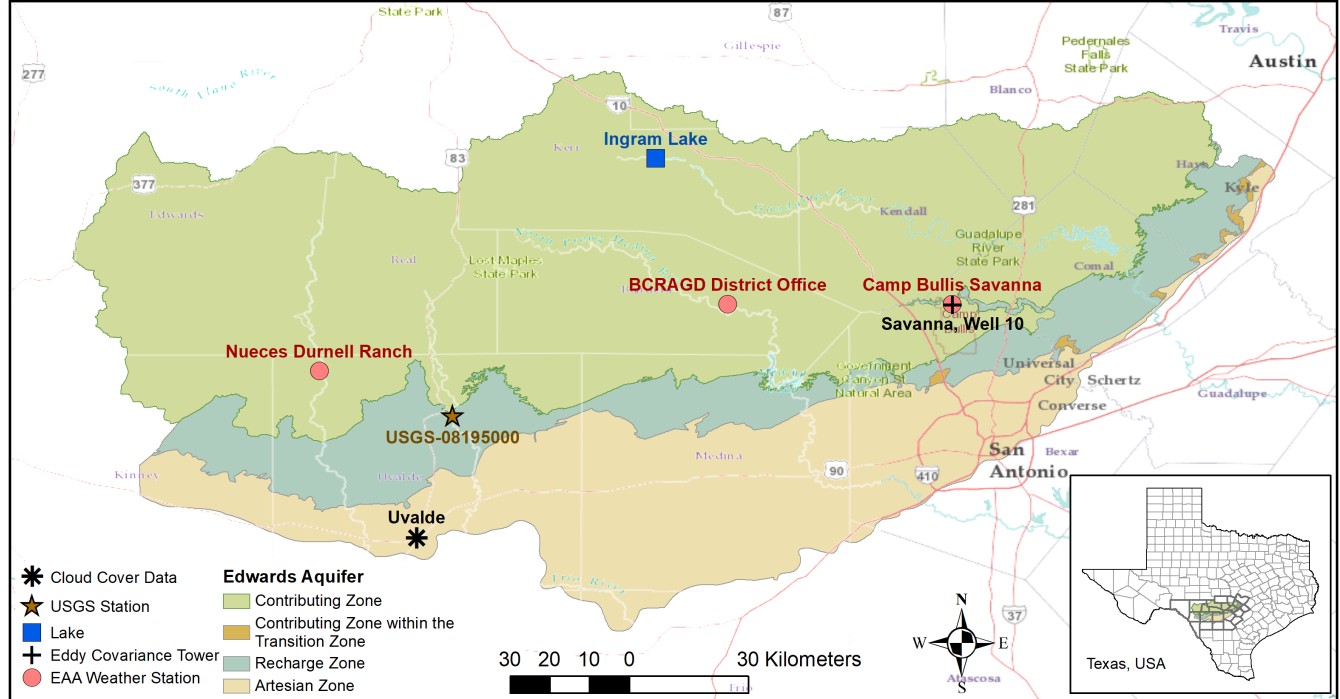

**Figure 2.** Data source locations across the Edwards aquifer region. The map shows the location of EAA's weather stations with local climate data, the U.S. Geological Survey (USGS)'s station with surface water temperature data, Ingram Lake with estimated lake evaporation data, Uvalde city with the cloud cover data, and the eddy covariance tower with the actual evapotranspiration data. BCRAGD refers to the Bandera County River Authority and Groundwater District's office.

velopment Board (TWDB). 1.9% of these measurements were missing, which were filled in by linear interpolation. These measurements were upscaled to daily lake evaporation totals ($E_{sw}$) using monthly-varying pan coefficients developed by the TWDB. However, sporadically extremely high and low $E_{sw}$ values, shown in Fig. 4a, were found to be quantitatively incon-sistent with the climatic data ($T_a$, $R_s$, and $RH$) trends at the BCRA station, provided in Appendix A. Therefore, this time

series is regarded as anomalous. Such anomalies are quite common in $E_p$ measurements due to birds drinking from the pan, debris falling in, or water splashing out (Thompson, 1999). Subsequently, these anomalies are carried into the daily $E_{sw}$ data, but largely smoothed out in monthly-averaged $E_{sw}$. Because the ML model was run with daily $E_{sw}$ data here, a 7-day rolling median function was used to reduce the noise and outliers in the daily $E_{sw}$ data (Fig. 4a). Monthly $E_{sw}$, derived from daily $E_{sw}$ (Fig. 4) were then used to determine the suitability of the MFs to predict the monthly $E_{sw}$ at Ingram Lake.

The daily and monthly MFs rely on surface water temperature, $T_{sw}$, rather than $T_a$, in $e^o$ and $e_a$ calculations (Eqs. 6-7). The closest gauging station, with the surface water temperature data from 9/1/2015 to 12/31/2019 at the 15-min (or 1-hr) intervals, to Ingram Lake is the U.S. Geological Survey Station (USGS 08195000) located at the Frio River in Concan, TX. The Frio River at the USGS 08195000 and Ingram Lake are small-size surface water bodies fed by groundwater from the Trinity aquifer.



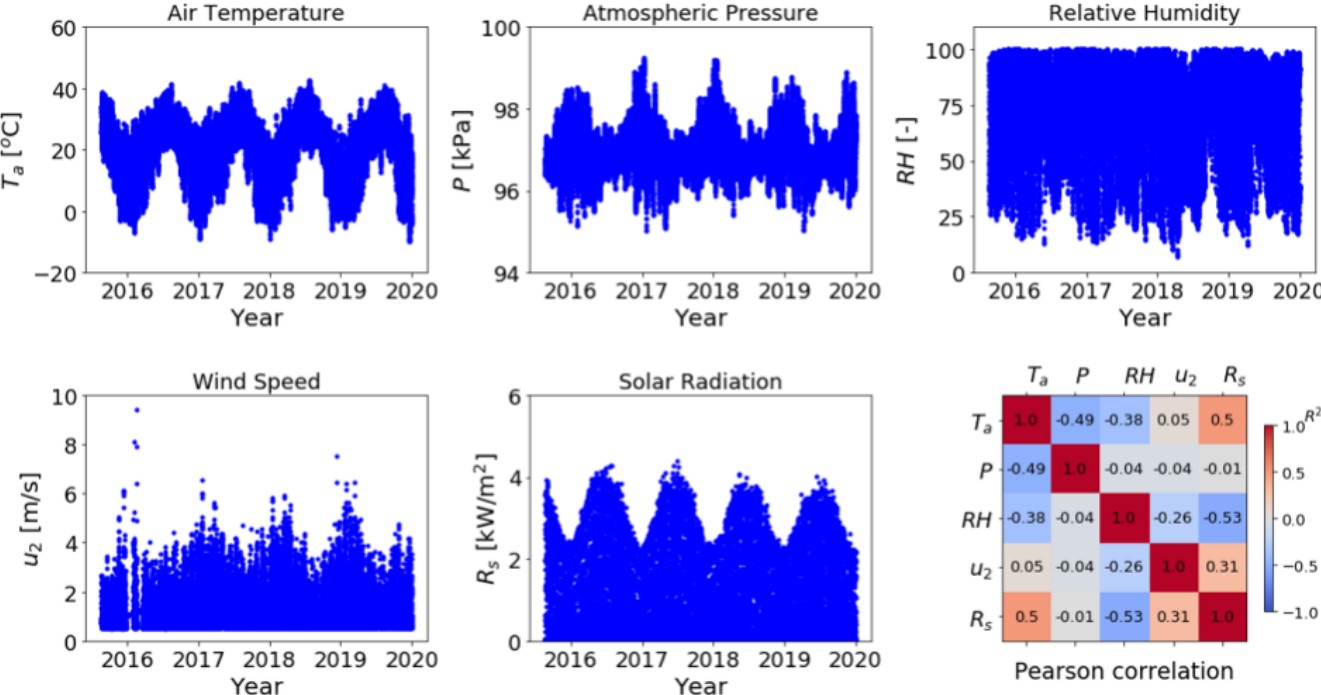

**Figure 3.** Hourly climate data and statistical correlations among them at the NDR weather station.

Therefore, $T_{sw}$ from the Frio river were used in MF-based $E_{sw}$ calculations at Ingram Lake. Daily-averaged $T_{sw}$ are shown in

Fig. 4b. Because the Frio river is a groundwater-fed river, $T_{sw} \geq T_a$ in winter; whereas, $T_{sw} \leq T_a$ in summer. 0.26% of daily $T_{sw}$ were missing, which were filled in by linear interpolation.

**Actual Evapotranspiration Measurements.** $ET_a$ measurements were obtained from the EC tower at Savanna, Well 10 near Camp Bullis, TX. Instruments were installed approximately 1.2 m above the height of the vegetation. Vegetation at the EC tower is open oak savanna. Daily $ET_a$ data were available from 5/4/2016 to 1/21/2019 (Appendix A). 2 ($< 0.1\%$) daily $ET_a$

measurements were missing, which were filled in by linear interpolation.

## 3 Results and Discussion

**Potential Evapotranspiration via Penman-Monteith Equation.** Hourly $ET_o$ were calculated via Eq.1, using local climate data at the NDR weather station from 9/1/2015 to 12/31/2019. In Fig. 5a, $R_a$, computed using Eq. 4, relies on information on the geographic location of the weather station and hourly-varying solar time angle. $R_a$ was subsequently used to calculate $R_{so}$.

The ratio of the measured $R_s$ to the computed $R_{so}$ provides an estimate for cloudiness, defined as the fraction of the number of cloudy-sky hours in a day. In Fig. 5b, PME-computed monthly-averaged cloudiness from 2016 through 2019 agrees well with the monthly-averaged representative cloudiness for the Uvalde city.





(a) Lake evaporation

(b) Surface water and air temperature

**Figure 4.** Surface water measurements closest to the NDR weather station. Surface water temperatures at 15-min intervals were obtained from Frio river in Concan, and daily lake evaporation data were obtained from Ingram Lake in Texas.

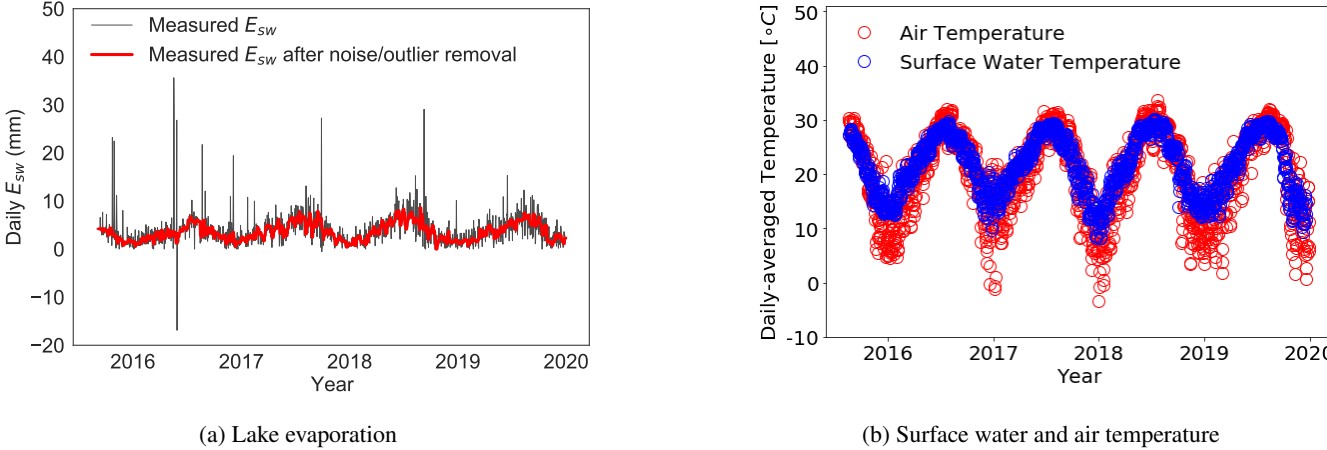

(a) Extraterrestrial radiation

(b) Monthly cloud coverage in 2016 - 2019

(c) Longwave (outgoing) radiation

(d) Potential evapotranspiration

**Figure 5.** Intermediate results (a) - (c), and daily or monthly potential evapotranspiration totals (d) at the NDR station computed by Eq. 1.





Measured $R_s$ was the input to the PME. $R_{nl}$ in Fig. 5c was computed by Eq. 2, based on daily extremes of air temperature, and hourly cloud cover and actual vapor pressure. $R_{nl}$ was then used to calculate the net solar radiation, $R_n = R_{nl} - R_s$,

in Eq.1. Hourly $ET_o$ were aggregated to daily $ET_o$, which is the time interval at which both $E_{sw}$ and $ET_a$ measurements were available. Although hourly $ET_o$ contains negative values at the humid and rainy hours, daily $ET_o$'s were persistently non-negative (Fig. 5d), as expected for the Texas climate.

**Lake Evaporation Using Meyer's Formula**. Daily and monthly $E_{sw}$ data from Ingram Lake, the closest water body to the NDR site, have been reported by the TWDB. Applicability of Eqs. 6 and 7 to predict monthly $E_{sw}$ were tested here using

monthly $E_{sw}$ data. In this analysis, $E_{sw}$ at Ingram Lake computed using Eq. 6 averaged over each month to obtain monthly $E_{sw}$. Local climate data from two weather stations, the NDR station ($\sim 90$ km away) and the BCRA station ($\sim 40$ km away), were used in calculations.

In Fig. 6a, monthly-averaged daily $E_{sw}$ computed by Eq. 6 matched the $E_{sw}$ data almost perfectly ($R^2 = 0.99$) when the climate data at the NDR station was used, but underpredicted the $E_{sw}$ when the climate data at the BCRA station was used. As

shown In Fig. 6b, the original form of Eq. 7 with $C = 1$ matched the overall monthly trend of the $E_{sw}$ data, but underestimated the magnitude of $E_{sw}$ irrespective of climate data from the NDR or BCRA station. When $C = 1.6$ with the BCRA data (and $C = 1.5$ with the NDR data), Eq. 7 matched the monthly $E_{sw}$ data almost perfectly ($R^2 = 0.99$). Although the empirical relations in Eqs. 6 and 7 were independently derived using site-specific data at two markedly different geographic locations in Canada and England, these equations matched monthly $E_{sw}$ in south-central Texas surprisingly closely in Fig. 6.

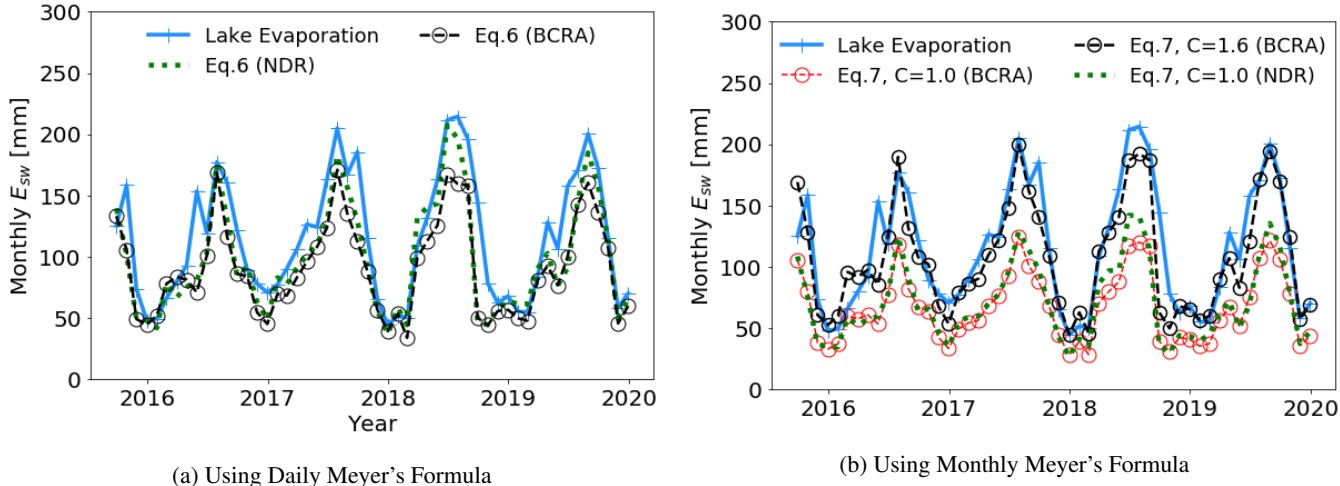

(a) Using Daily Meyer's Formula      (b) Using Monthly Meyer's Formula

**Figure 6.** Comparison of surface water evaporation at Ingram Lake computed by using Meyer's Formula (MF) and local climate data at the NDR and BCRA weather stations against TWDB's lake evaporation data.

**Computed Potential Evapotranspiration vs. Lake Evaporation**. Fig. 7 shows that $ET_o$, in general, set the lower bound for $E_{sw}$ at Ingram Lake for the entire period. In 2016, $ET_o \sim E_{sw}$ for most of the year except in December. Although $E_{sw} > ET_o$





in the summer of the following years, with the largest difference in the summer of 2018, $ET_o$ appears to be a reliable predictor for $E_{sw}$ especially in spring and winter months.

Fig. 7 reveals that $ET_o$ computed by the PME using the climate data from the NDR or BCRA stations can be used to estimate
the minimum monthly $E_{sw}$, computed by MF and determined by TWDB, for Ingram Lake. This is rather an interesting result, given that the empirical MF does not involve complex solar radiation (e.g., extra-terrestrial, clear-sky, outgoing longwave) calculations as in the PME and relies on surface water temperatures rather than air temperatures. It can be argued that surface water temperature may implicitly account for the solar radiation effect on $E_{sw}$. Nevertheless, the main conclusion from Fig. 7 is that $ET_o$ at the NDR station set the lower bound for $E_{sw}$ at nearby Ingram Lake (i.e., $ET_o \leq E_{sw}$). Fig. 7 further suggests
that if uncertainty in local climate measurements are higher than lake evaporation measurements, mathematically simpler MF, after being validated with historical lake water evaporation data, can be used to predict potential evapotranspiration from new lake evaporation data. Fig. 7 also shows that monthly-aggregated $ET_o$ near Ingram Lake is slightly higher in summers than at the NDR or BCGA sites. However, if the climate data near Ingram Lake is not available, data from a farther weather station to the west can be used to predict the $ET_o$, and hence, the lower bound for $E_{sw}$. Based on the existing data, the results also
suggest that no additional weather station is needed between the NDR and BCGA stations to predict $ET_o$ and/or $E_{sw}$ from other surface water bodies fed by the same groundwater system between these two stations.

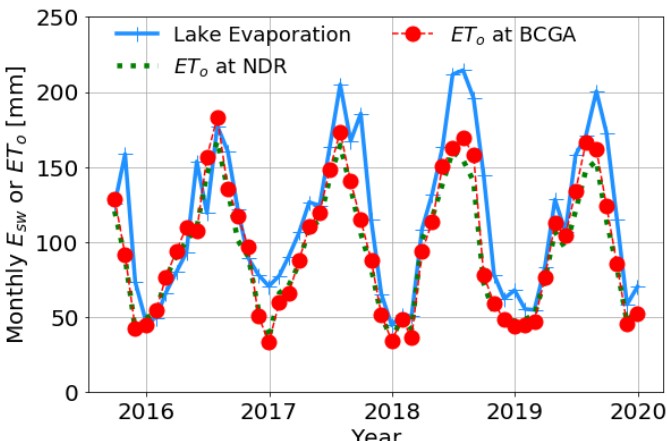

**Figure 7.** Comparison of monthly lake water evaporation against computed monthly potential evapotranspiration using local climate data from the NDR and the BCGA weather stations.

**Potential Evapotranspiration vs. Actual Evapotranspiration**.

Fig. 8 compares daily or monthly Bowen-ratio-corrected $ET_a$ measurements from the EC tower against $ET_o$ computed by the PME, using local climate data from the CBS station. According to this plot, $ET_o \geq ET_a$ during the monitoring period, as
expected. In some summer months, $ET_o$ was about three times higher than $ET_a$ (e.g., July 2017), indicating that the soil was





too dry in summer times to contribute to the evapotranspiration at the CBS site. As compared to $ET_o$ at the NDR site, $ET_o$ is typically higher at the CBS site in summer months, revealing spatial variability in $ET_o$. The results in Figs. 7 and 8 lead to

$$ET_a \leq ET_o \leq E_{sw}. \tag{8}$$

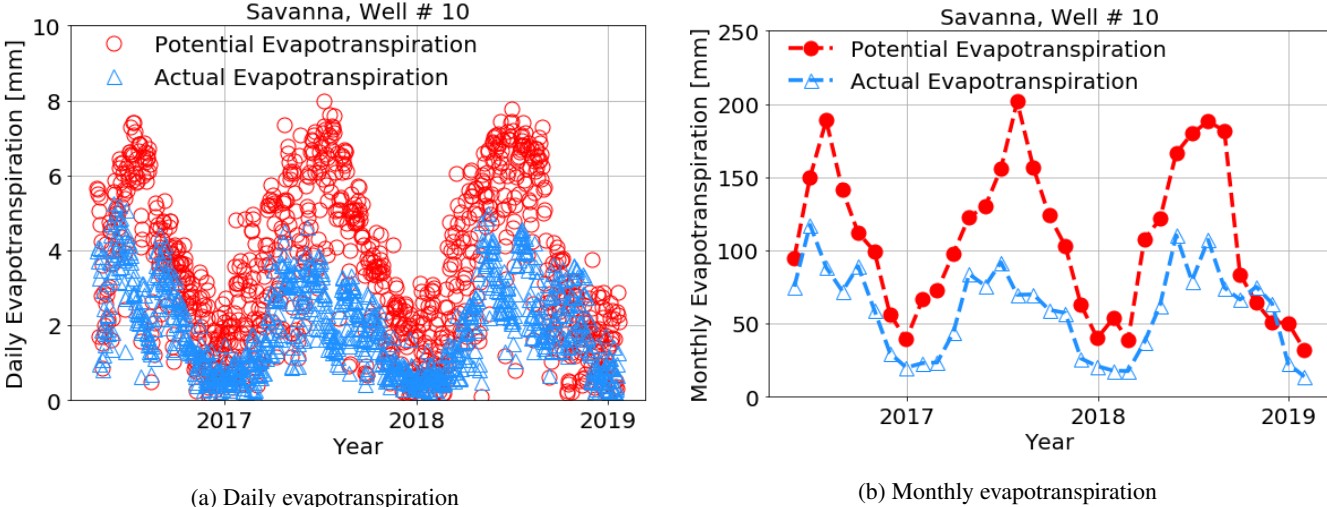

(a) Daily evapotranspiration

(b) Monthly evapotranspiration

**Figure 8.** Daily or monthly measured actual evapotranspiration from the EC tower at the Camp Bullis site vs. potential evapotranspiration computed by Eq. 1

$ET_a$ is the most critical evapotranspiration estimate, especially for irrigation, agricultural, and water resources management
practices. As discussed previously, the EC method provides the most accurate prediction for $ET_a$; however, the associated capital and maintenance costs are high (e.g., the capital cost for the EC tower at the CBS site was about \$40,000 and required frequent maintenance). Thus, EC-based $ET_a$ data acquisition is expensive. On the other end, $E_{sw}$ measurements are important indicators of global climate change (Wang et al., 2018), which could affect the water levels & chemistry, and the sustainability of the lake habitat. Existing monthly pan evaporation coefficients, however, have inherent uncertainties and their potential
adjustments for future climate conditions remain unclear. In brief, capital and operational costs for $ET_a$ measurements, and the accuracy of the upscaling method to determine $E_{sw}$ for the current & future climate conditions are the main challenges that practitioners face.

Considering the aforementioned challenges, we present a robust ML model using the local climate data as the independent feature that can (i) predict $ET_o$ as an alternative to computationally intensive PME; (ii) predict $E_{sw}$ to eliminate uncertainties
associated with pan evaporation measurements and pan evaporation coefficients needed to upscale $E_p$ to $E_{sw}$; (iii) predict $ET_a$ to offset the high capital and operational costs for EC towers. In addition, we explain the nonlinear feature dependencies on the $ET_o$, $E_{sw}$, & $ET_a$ predictions, based on solid game theory, to enhance the transparency and interpretability of the ML model.




**Predictive ML Models**. We investigated if daily $ET_o$ can be accurately computed by the probabilistic ML model using local climate data, as an alternative to Eq. 1. The ML model was trained by using $90\%$ of the daily climate data & the month of the

year as features, and the PME-computed $ET_o$ data as the target. Subsequently, the trained ML model was used to predict $ET_o$ for the remaining $10\%$ of the data (testing period). In the end, ML-predicted daily $ET_o$ were compared against PME-computed daily $ET_o$ to assess the performance of the ML-model on the testing data. Differences between the ML-predicted $ET_o$ from the PME-computed $ET_o$ on the testing dataset are shown in Fig. 9a, in which $\sim 99\%$ of the PME-computed $ET_o$ were within the model's $95\%$ prediction interval. In other words, the model was successful $\sim 99\%$ of the times in determining the exact

interval around each predicted ($ET_o$) value such that there is a $95\%$ probability that the corresponding target ($ET_o$) value is within this interval. Additionally, based on the statistical measures in Table. 1, calculated using the point predictions from the model, the $ET_o$ predictions by the hybrid NGBoost-XGBoost ML model can be used as a reliable alternative method to estimate watershed-scale $ET_o$. The total training time for the $ET_o$ hybrid model was $\sim 30$ minutes that involved choosing the optimum model out of 230 candidates using a 3-fold grid search cross-validation technique, which equates to 690 model fits

on an Intel Core i9-9980XE CPU and 64 GB RAM computer. The main advantage of the ML-based $ET_o$ prediction model is that it does not require computationally-involved extra-terrestrial, clear-sky, and outgoing longwave (outgoing) solar radiation, as part of net solar radiation calculations.

**Table 1.** ML predictive model accuracy test with statistical measures.

|  | Data | RMSE*(mm) | MAE$^{\dagger}$(mm) | $R^{2}$ $^{\ddagger}$ |
|---|---|---|---|---|
| $ET_o$ | Training data only | 0.099 | 0.074 | 0.996 |
|  | Testing data only | 0.139 | 0.102 | 0.992 |
| $E_{sw}$ | Training data only | 0.703 | 0.545 | 0.843 |
|  | Testing data only | 0.918 | 0.736 | 0.750 |
| $ET_a$ | Training data only | 0.388 | 0.291 | 0.891 |
|  | Testing data only | 0.533 | 0.411 | 0.804 |

(*) Root mean square error; † Mean absolute error; ‡ Pearson correlation

The ML-based $E_{sw}$ prediction model was trained by using the first $90\%$ of the daily climate data & the month of the year as features, and the measured $E_{sw}$ data as the target. Subsequently, the model was tested on the remaining $10\%$ of the data.

The comparison between ML-predicted daily $E_{sw}$ and the measured $E_{sw}$ on the testing data is shown in Fig. 9b. The ML-predicted $E_{sw}$ matched the measured $E_{sw}$ very closely, and $\sim 90\%$ of the actual $E_{sw}$ were within the model's $95\%$ prediction interval in the testing dataset. Based on the statistical measures in Table. 1, probabilistic prediction of $E_{sw}$ by the hybrid NGBoost-XGBoost ML model can be used as a reliable method for $E_{sw}$ projections. The total training time for the $E_{sw}$ hybrid model was $\sim 6$ minutes, including choosing the optimum model from 690 model fits. The main advantage of the ML-based

$E_{sw}$ prediction model is that $E_{sw}$ predictions are not affected by anomalies in $E_p$ measurements (Fig. 4a) or uncertainties in monthly pan evaporation coefficients.





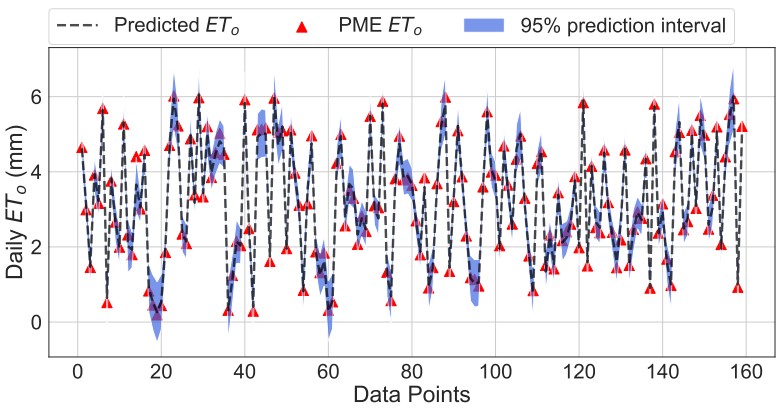

(a) $ET_o$ predictions on the testing data

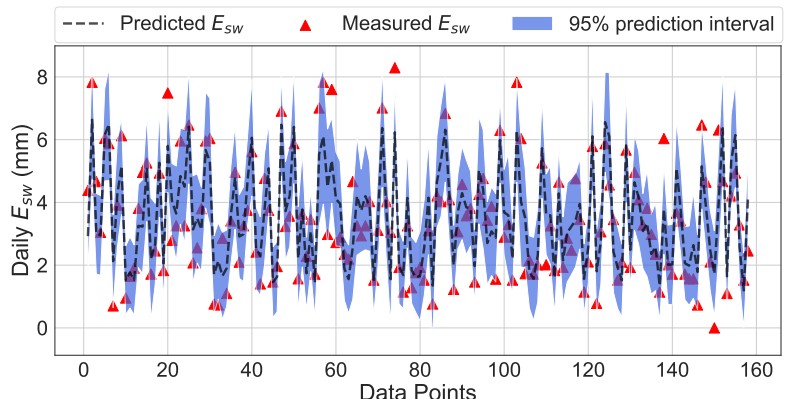

(b) $E_{sw}$ predictions on the testing data

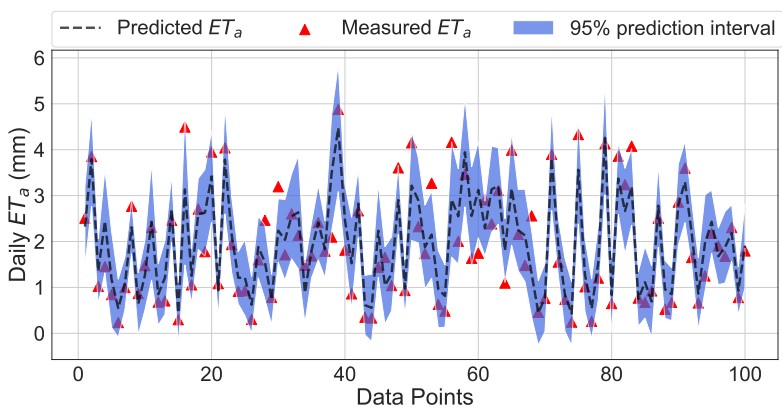

(c) $ET_a$ predictions on the testing data

**Figure 9.** Graphical representation of the predictive capability of the hybrid NGBoost-XGBoost model.





The ML-based $ET_a$ prediction model was trained by using the first $90\%$ of the daily $R_s$, PME-computed $ET_o$, & the month of the year as features, and the measured $ET_a$ data as the target. Subsequently, the model was tested on the remaining $10\%$ of the data. The comparison between ML-predicted daily $ET_a$ and the actual $ET_a$ measurements - on the testing data - is

shown in Fig. 9c, in which $\sim 91\%$ of the actual $ET_a$ values were found within the model's $95\%$ prediction interval. ML-based $ET_o$ predictions was more accurate than the $ET_a$ predictions due, in part, to the availability of less data from the EC tower at the CBS site than at Ingram Lake or the NDR site for the ML-model training. However, based on the statistical measures in Table. 1, probabilistic prediction of $ET_a$ by the hybrid NGBoost-XGBoost ML model, with an $R^2$ of 0.804 on testing data, can still be used as a reliable method to estimate the future $ET_a$. The total training time for the $ET_a$ hybrid model was $\sim 9$

minutes, including the 690 model fits to choose the optimum model. The main advantage of the ML-based $ET_a$ prediction model is that it offsets the high capital and maintenance costs for the installation and operation of EC towers to acquire $ET_a$ measurements. Obviously, if more $E_{sw}$ and $ET_a$ measurements are available for ML-model training, the predictive accuracy of the respective ML models would improve. The EAA is planning to construct additional EC towers in other parts of the aquifer region to collect more $ET_a$ data, which would enhance the training and predictive performance of the ML-model. Similarly, as

the TWDB continues to collect $E_{sw}$ data by effectively filtering out observed anomalies, availability of longer $E_{sw}$ data with less noise for ML-model training would enhance the predictive accuracy of the model.

**Feature Importance in** $ET_o$, $E_{sw}$**, and** $ET_a$ **Predictive ML Models**. It is imperative for end-users to peek inside ML models to better understand how the features contribute to the model predictions or how they affect the overall model performance. To this end, we investigated the relationship and contribution of each feature to the prediction of the $ET_o$, $E_{sw}$, and $ET_a$

models using Shapley values – a method from coalition game theory. The Shapley value is the average marginal contribution of each feature value across all possible combinations of features. The features with large absolute Shapley values are deemed important. To obtain the global feature importance, we average the absolute Shapley values for every feature across the data, sort them in decreasing importance and plot them. Each point on the plot represents a Shapley value for individual features and instances. The position on the x- & the y-axis is determined by the Shapley values & the feature importance, respectively, and

the color scale depicts the feature importance from low to high. Interested readers may refer to Lundberg et al. (2020) for the mathematical and algorithmic background of the Shapley value calculations.

Fig. 10a shows that the order of importance of local climate variables from the highest to the lowest on the computed $ET_o$ involves the $R_s$, $T_a$, $RH$, $u_2$, & $P$. The month of the year is deemed to be the second least important feature for the $ET_o$ model. This finding is important to evaluate the suitability of the simplified versions of the PME proposed for semi-arid watersheds

with scarce climate data. Irmak et al. (2003) proposed two simplified PMEs that require less number of climate variables to calculate the net radiation ($R_n$ in Eq. 1). The first equation relied on the measured $T_a$ and $R_s$, whereas the second equation relied on predicted $R_s$, and measured $T_a$ and $RH$. Although the simplified equations were used to estimate $R_n$ only, the second equation, built on the three most important climate variables identified in Fig. 10a for more accurate $ET_o$ estimates, is expected to perform better for the semi-arid regions, if the predicted $R_s$ has low uncertainty. This is consistent with the conclusion by

Irmak et al. that the second equation accounted for $79\%$ of the variability in $R_n$ in their case studies.





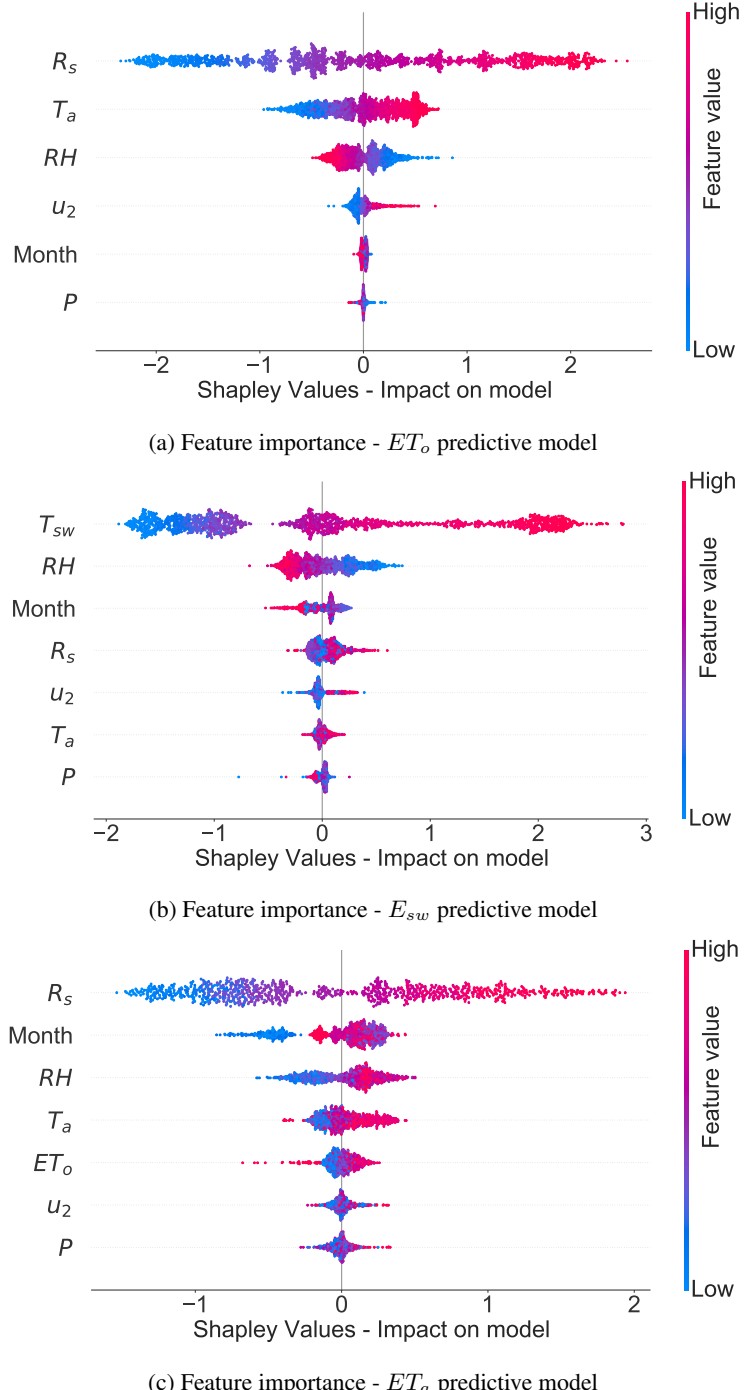

(a) Feature importance - $ET_o$ predictive model

(b) Feature importance - $E_{sw}$ predictive model

(c) Feature importance - $ET_a$ predictive model

**Figure 10.** The order of importance of climate variables on the $ET_o$, $E_{sw}$, and $ET_a$ predictions.





$R_s$ was recently used as a surrogate variable to reduce the uncertainty of $ET_o$ projection data (Yoo et al., 2020). This can be justified by the findings from Fig. 10a, in which $R_s$ displayed a more profound impact on $ET_o$ than the other forcing variables. On the other hand, the mean annual temperature was used by Hartmann et al. (2017) as a proxy for $ET_o$ in assessing aquifer recharge sensitivity to climate variability based on the argument that $R_n$ is temperature-dependent and temperature is the

best-understood and most common climatic variable for large-scale hydrological models. Similarly, a computationally simple method of Berti et al. (2017) that relies only on $T_a$ was reported to be the best alternative method to the PME in describing spatiotemporal characteristics of $ET_o$ in different sub-regions of mainland China (Peng and Feng, 2017). Such assumptions (Hartmann et al., 2017) and conclusions (Peng and Feng, 2017), however, should be made with caution in $ET_o$ calculations, especially for semi-arid regions, as the ML analysis unveiled that $R_s$ (as part of $R_n$ in Eq. 1) is more important than $T_a$ in $ET_o$

prediction. Moreover, Figs. 3 and 11 revealed that the statistical correlation between $R_s$ and $T_a$ is weak with $R^2 \le 0.6$. Thus, the use of $T_a$ as a proxy for $ET_o$ is questionable for the semi-arid regions.

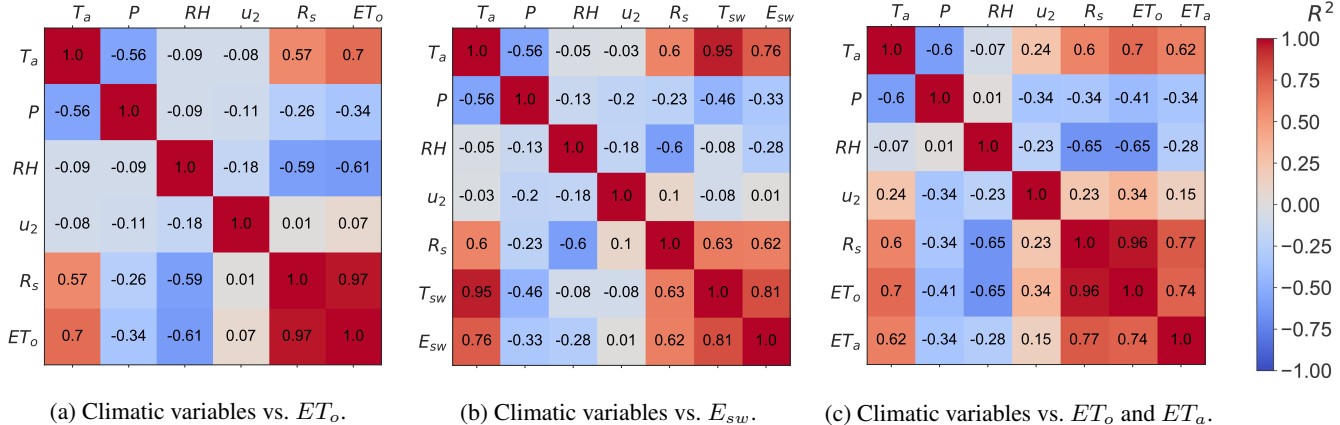

(a) Climatic variables vs. $ET_o$.  (b) Climatic variables vs. $E_{sw}$.  (c) Climatic variables vs. $ET_o$ and $ET_a$.

**Figure 11.** Correlation map between daily climatic variables and (a) the potential evapotranspiration at the NDR site, (b) lake evaporation at Ingram Lake, and (c) and actual evapotranspiration at the Camp Bullis site.

Gong et al. (2006) noted that although the order of importance of climate variables on $ET_o$ estimates through the PME varied with season and region in their study, $ET_o$ in general was most sensitive to $RH$, followed by $R_s$, $T_a$ and $u_2$. The authors used time-histories of daily $T_a$, $u_2$, $RH$, and daily sunshine duration. In our analysis, however, measured climate variables were

available at the 15-min intervals, including also $R_s$ and $P$. Unlike the general conclusion by Gong et al., our ML-based feature importance calculations in Fig. 10a revealed that both $R_s$ and $T_a$ were more critical than $RH$ on $ET_o$ estimates.

Fig. 10b shows that the $E_{sw}$ is largely impacted by the $T_{sw}$ followed by $RH$. $T_a$ is given lower importance because of the high correlation ($R^2 = 0.95$) between $T_{sw}$ and $T_a$ (Fig. 11b), and thus, the model considers $T_a$ as redundant. Fig. 10b also highlights the model's understanding of the underlying hydrological process. For example, we see that the model tries to push

the $E_{sw}$ predictions upward - represented by higher Shapley value on the x-axis - when the $T_{sw}$ feature values are high - represented as red dots - and the $RH$ feature values are low - represented as blue dots. In other words, after being trained with





the historical data, the ML model predicts higher $E_{sw}$ when $T_a$ is higher and $RH$ is lower, representing the underlying physics correctly, and hence, evidencing of its capability of making learning-based conscious predictions.

For the $ET_a$ predictions, we found that the $R_s$ followed by the month of the year, $RH$, and $T_a$ are the most important factors
for $ET_a$ predictions, as shown in Fig. 10c. However, in comparison to $ET_o$, $ET_a$ is dependent less strongly on $R_s$ ($R^2 = 0.77$), as the time-dependent soil moisture and vegetation transpiration also impact $ET_a$ measurements, unlike for PME-computed $ET_o$ in Eq. 1. Besides, the analysis did not reveal a strong impact of $ET_o$ on the $ET_a$ predictions, because $ET_o$ calculations are based on the assumption of a hypothetical reference crop growing in a saturated soil (Section 2), and hence, not accounting for the effect of temporally-varying transpiration rates from the actual vegetation (open oak savanna at the EC tower site) and
the transient nature of the soil moisture content affecting $ET_a$ measurements. Due to the temporal variations in water uptake, vegetation transpiration, and soil moisture content on the field near the EC tower, where $ET_a$ measurements were taken, the month of the year became a strikingly more important feature in the $ET_a$ model than in the $ET_o$ model. Interestingly, we found many instances where the ML model tries to push the $ET_a$ predictions higher - represented by higher Shapley value on the x-axis - when the $RH$ is relatively high - represented as red dots - in Fig. 10c. In other words, the ML model in certain conditions
predicts higher $ET_a$ when $RH$ is high. Such findings were also reported by Yan and Shugart (2010) from $ET_a$ measurements by the EC method. High $ET_a$ at high $RH$ could be attributed, for example, to high air-vapor uptake by water deficit soil and vegetation in hot and humid days, which are subsequently released back into air due to evaporation from soil and transpiration from vegetation; or evaporation from saturated soil and transpiration from vegetation in high $RH$ conditions following rain events; or evaporation from moist soil on a cold day following rain events. Unlike the ML-based modeling, the dynamics
between soil moisture, vegetation water uptake, rain events, $T_a$, $RH$ and $ET_a$ cannot be captured by one-to-one correlation, as shown in Fig. 11. Additionally, Fig. 12 shows that, in certain situations, the model generates low $ET_a$ predictions despite high $ET_o$ & low $RH$ measures, which could be driven by critical moisture deficiency in the soil, especially in hot and dry summer. This could be a concern in future, as for a 2°C of global warming, most of Texas was projected to experience more than a doubling in the number of days above 38°C (Wobus et al., 2018). Such more frequent high $T_a$ over extended periods
could increase the soil moisture deficiency, and decrease aquifer recharge and springs flow, which could affect sustainability of groundwater for consumptive water uses and environmental flows for groundwater-bound threatened and endangered species.

## 4 Conclusions

Reliable prediction of actual evapotranspiration, $ET_a$, is useful to determine aquifer recharge in semi-arid regions, which is crucial for development of groundwater management plans for sustainable consumptive water use while maintaining quality and
quantity environmental flows to protect delicate habitats for groundwater-dependent species under current and projected climatic conditions. Similarly, reliable lake evaporation, $E_{sw}$, estimates and projections are important for lake water management, recreation, infrastructures on the lake shore, lake habitat, local climate change, and water cycle. Potential evapotranspiration, $ET_o$, are often used to determine the climate-driven watershed-scale evaporation power of the atmosphere, which could be extended to $ET_a$ predictions, if it is coupled with crop and soil information, and a surface conductance model.





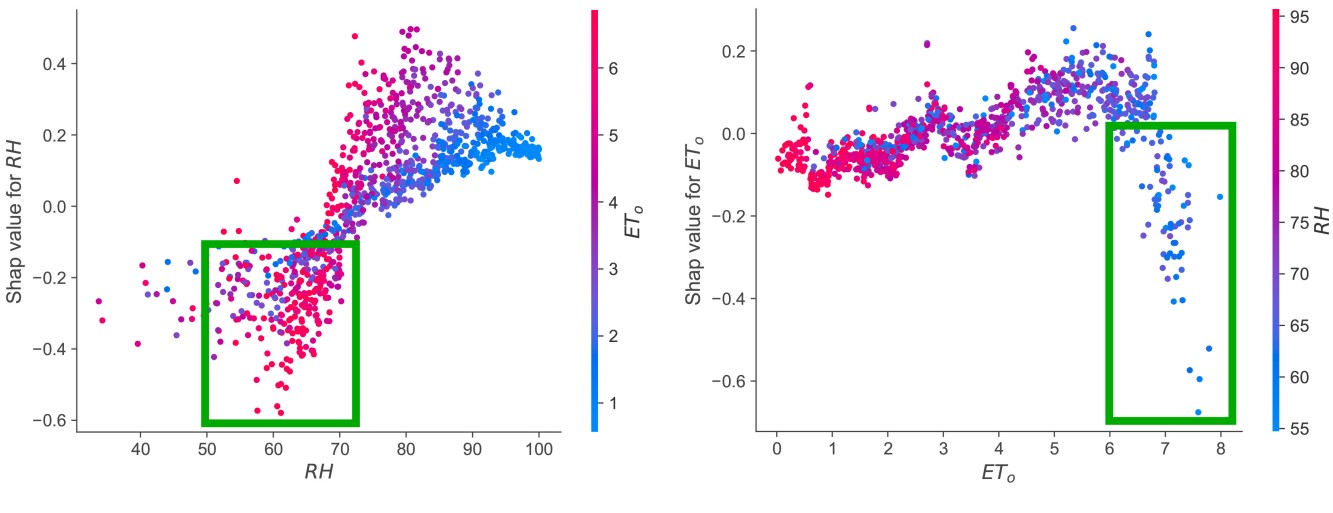

(a) $RH$ dependence plot with $ET_o$ interaction.

(b) $ET_o$ dependence plot with $RH$ interaction.

**Figure 12.** Dependence & interaction plots revealing the interrelationship between $RH$, $ET_0$, and the corresponding Shap values. The Shap values represent the model's behavior to either push the $ET_a$ value higher or lower. A higher Shap value means that the model is trying to produce a higher $ET_a$ prediction, and vice-versa. The green boxes highlight the regions where low $RH$ values correspond to high $ET_o$ values but low $ET_a$ predictions, which could be attributed to soil moisture deficiency.

Eddy covariance (EC) towers provide accurate estimates for site-specific $ET_a$, but at the expense of high capital, operational, and maintenance costs, which often limit the use of multiple EC towers and/or their operational periods in resources management projects. Pan evaporation method, on the other hand, is a simple, inexpensive, and widely-used data acquisition method to predict $E_{sw}$ at open water bodies, but suffers from uncertainties in pan evaporation measurements and in pan coefficients for the existing or projected climatic conditions when water evaporation is upscaled from the pan-scale to the large open-

water body-scale. $ET_o$ is often computed by energy-balance models, such as Penman-Monteith equation (PME) that relies on time-series of more local climate variables and includes rather complex calculations for net solar radiation computations. In brief, $ET_a$ measurements are often challenged by the project budget; whereas, $E_{sw}$ measurements are affected by uncertainties in and upscaling of pan-evaporation measurements. $ET_o$ calculations, on the other hand, require computationally-involved calculations.

To eliminate these shortcomings in $ET_a$, $E_{sw}$, and $ET_o$ predictions, we proposed a hybrid ML *probabilistic prediction* model of $ET_o$, $E_{sw}$, and $ET_a$ using the local climate data and the month of the year as the only independent feature. Different from other ML models, the proposed hybrid ML model is able to produce point predictions as well as a probability distribution over the entire outcome space for quantifying the uncertainties related to hydrological predictions. The proposed hybrid model could provide practitioners with a better understanding of the uncertainty in the $ET_o$, $E_{sw}$, and $ET_a$ predictions without

compromising the accuracy of the predictions. Our results showed that the hybrid (NGBoost-XGBoost) ML model successfully predicted the PME-computed $ET_o$, and the measured $E_{sw}$ and $ET_a$, in which $\geq 90\%$ of the target data points were within



the 95% prediction interval of the model, and $R^2$ values for the point predictions were 0.99, 0.75, and 0.8, respectively, using data from the model testing period. These results exhibit that the proposed hybrid ML model is a reliable and robust alternative method to predict $ET_o$, $E_{sw}$, and $ET_a$ from local climate data, without implementing computationally-intensive
PME calculations, or coping with uncertainties in $E_{sw}$ estimates using evaporation pans, or having expensive EC tower setups for $ET_a$ measurements.

We also demonstrated that the hybrid ML model, based on a game theory approach, generated new knowledge, different from sensitivity analyses findings in the existing literature, about the importance of features (variables) on the $ET_o$, $E_{sw}$, and $ET_a$ predictions. The underlying idea behind this analysis was to explain the prediction of an instance by computing the contribution
of each feature to the prediction. Our analysis revealed that the shortwave solar radiation, air temperature, and relative humidity are the most critical features for the $ET_o$ predictions, whereas the surface water temperature, relative humidity, and the month are the most critical features for the $E_{sw}$ predictions, and the shortwave solar radiation, month, and relative humidity are the most critical features for the $ET_a$ predictions in the semi-arid climate.

The EAA has 12 active weather stations across the Edwards aquifer region. The proposed hybrid ML models would allow
continuous $ET_a$ predictions, without the need for expensive EC tower setups, from continuously streaming climate data at these weather stations and through their interpolation between the stations. Moreover, the ML model would allow $E_{sw}$ predictions from surface water bodies without equipped with sensors or tools for $E_{sw}$ measurements, if they are located closer to the weather stations. Thus, the ML-model would curtail data acquisition costs and ML-based $ET_a$ predictions would be particularly useful for real-time aquifer recharge estimates, and irrigation and agricultural water management.

The proposed hybrid ML model would also be a useful tool to project local evapotranspiration and its impacts on aquifer recharge when projected local climate data over the next 30-50 years is statistically-downscaled from global climate/regional climate data and reinterpreted as a time series of local climate data. Such projections are crucial to predict potential drought of records ahead of time before it results in irreversible damages on the sustainability of groundwater resources for diverse consumptive uses and habitats of groundwater-bound endangered or threatened aquatic species.

*Data availability.* Data are available from the authors.

**Appendix A**

**Local climate data at the BCRA:** The closest EAA's weather station to Lake Ingram is located at the BCRA. The local climate data at the BCRA station are available for the same period at the NDR station (Fig. A1). The total number of missing hourly data were 2 ($< 0.1\%$), which were filled by linear interpolation.



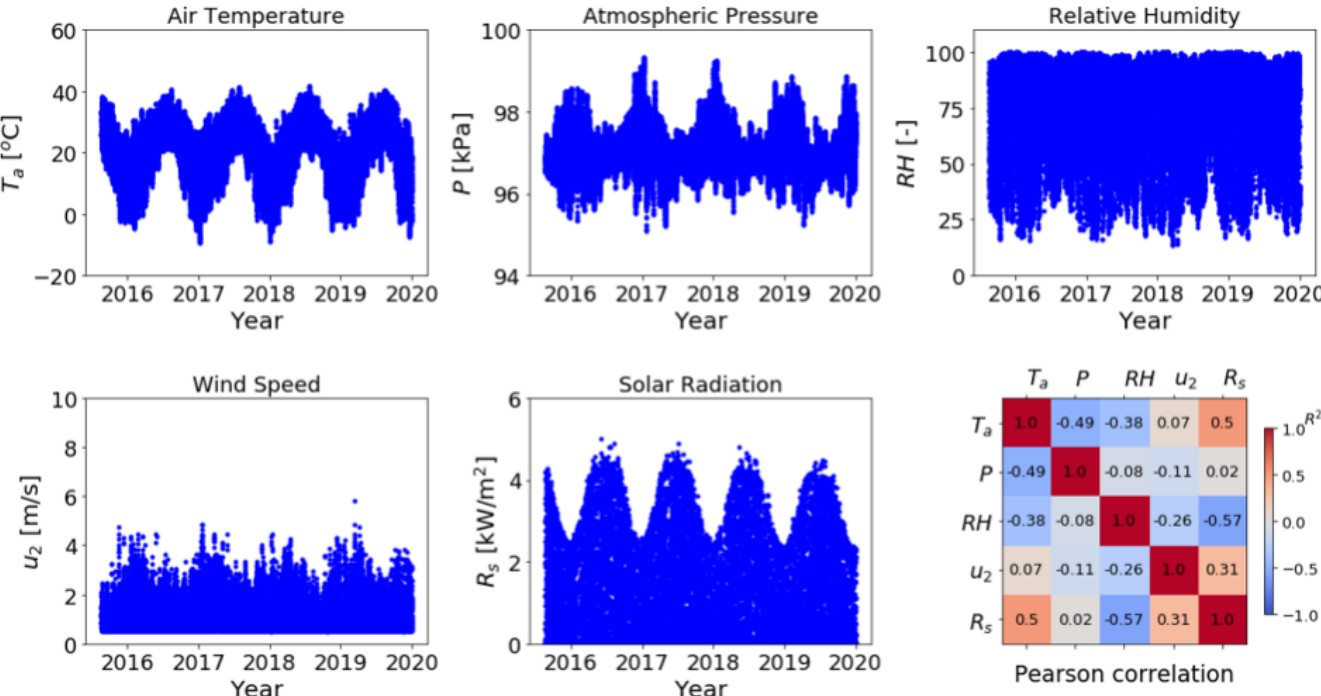

**Figure A1.** Historical hourly climate data at the EAA's BCRA weather station.

**Local climate data and $ET_a$ measurements at the Camp Bullis Site:** Daily $ET_a$ data were available from 5/4/2016 to 1/21/2019. 32 $(< 0.1\%)$ local climate data at 15-min intervals and 2 $(< 0.1\%)$ daily $ET_a$ measurements were missing during this period. Time histories of hourly-averaged $T_a$, $P$, $RH$, and $u_2$ and hourly-aggregated $R_s$ from the EAA weather station at Savanna, Well 10 near Camp Bullis, TX along with daily $ET_a$ measurements are shown in Fig. A2.

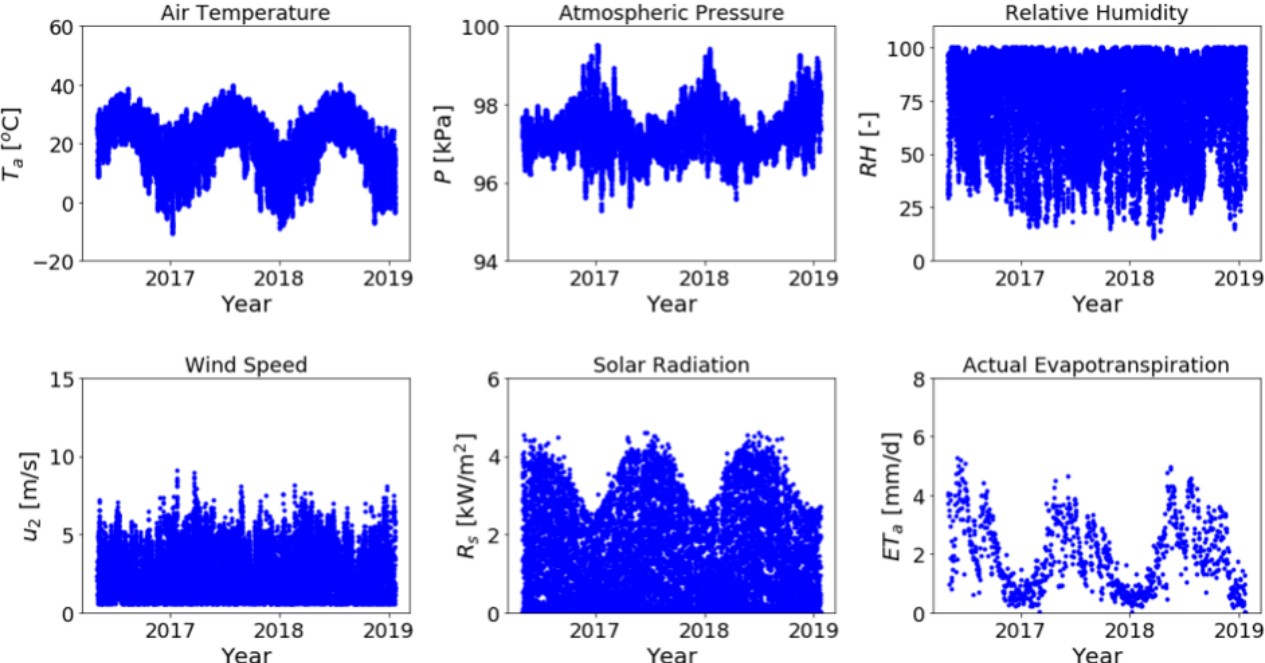

**Figure A2.** Historical hourly climate data from the EAA's weather station at Savanna, Well 10, and daily $ET_a$ measurements from the EC tower.

*Author contributions.* HB and JW performed database management, data quality checks, and PME and MF calculations. DC developed the
predictive ML models and performed the feature analysis. All authors are involved in conceptualization, analyzed the results, and wrote, reviewed, and edited the manuscript.

*Competing interests.* The authors declare that they have no conflict of interest.

*Acknowledgements.* The authors would like to thank to Newfel Mazari and Marcus Gary of EAA for their help with acquisition of climate data from EAA weather stations; Ned Throshanov and Sarah Eason of EAA for their help with preparation of the location map; John Zhu of
TWDB in Austin, TX for providing us with the raw daily pan evaporation data and follow-up discussions; and Tara Bongiovanni of BEG in Austin, TX for sharing with us the daily actual evapotranspiration data from the eddy covariance tower in Camp Bullis, TX.





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
