# Peer review of "A Hybridized NGBoost-XGBoost Framework for Robust Evaporation and Evapotranspiration Prediction"

_Hydrology and Earth System Sciences, 2020_

## Referee Comment (RC1) · Anonymous Referee #1 · 17 Aug 2020

In this work, the authors analyzed the relationship between potential evapotranspiration (ETo), actual evapotranspiration (ETa), and surface water evaporation, using data from multiple sources. Major comments:

1. Study objective #1 is not clear. Relationship between PET and Actual ET has been well studied in the literature. The authors should mention the previous works in this area. I found the literature survey is too cursory. Suggest that the authors move some materials from Section2 to the Introduction. Even so, it's not clear to me from the Intro why the existing results are not sufficient, such that the authors need a sophisticated ML to revisit the problem. The motivation needs to be elaborated further in the Intro.

[Figure]

2. Similarly, in Section 2 the motivation of using ML is not clear. What regression methods have been used before? Why existing methods are insufficient in terms of model performance? The authors need to touch on these aspects. Otherwise, the work seems to focus solely on a new ML algorithm without justification, and no baseline results (e.g., multivariate linear regression) were provided.

3. The ML pipeline is not clear. A diagram is needed to show inputs and output to the ML model. Around L185, the authors simply spelled out the inputs, without much reasoning. Why these features are selected? What is the lead time of prediction? The promise of ML is not so much for well gauged sites, but for sites with a lot of missing data.

4. Variable importance calculation is well established for tree-based method, which entails finding whether a variable is selected to split on during the tree building process, and how much the squared error (over all trees) is improved or reduced as a result. Why a new variable importance method is needed?

Minor comments:

In Abstract, the authors concluded "the deterministic analysis reveals that ETo set the upper bound for Eta", isn't this expected?

---

## Author Comment (AC1) · 28 Aug 2020

We thank Referee #1 for the questions & comments. We have interspersed our responses between the questions and comments. New texts, figures, Tables planned to be included in the manuscript in response to the comments are marked in blue in our responses below.
* * *
**Referee** #1:

In this work, the authors analyzed the relationship between potential evapotranspiration (ETo), actual evapotranspiration (ETa), and surface water evaporation, using data from multiple sources.

**Major comments: 1.** Study objective #1 is not clear. Relationship between PET and Actual ET has been well studied in the literature. The authors should mention the previous works in this area. I found the literature survey is too cursory. Suggest that the authors move some materials from Section 2 to the Introduction. Even so, it is not clear to me from the Intro why the existing results are not sufficient, such that the authors need a sophisticated ML to revisit the problem. The motivation needs to be elaborated further in the Intro.

**Response:**

*1.1 'objective #1 is not clear. Relationship between PET and Actual ET has been well studied in the literature'*

To be more accurate with the definition of different evapotranspiration processes in our revised manuscript in accordance with the description by Allen(1998) and McMohan et al. (2013) (both references were cited in the original manuscript), we have replaced 'PET' (referring to $ET_o$ in the original manuscript) with the 'reference crop evapotranspiration', and also provided the description of 'PET' and differentiated it from the reference crop evapotranspiration, $ET_o$. To that end, we plan to include the following description in the end of the first paragraph of the Introduction section:

Evapotranspiration can be computed as reference crop evapotranspiration ($ET_o$), actual evapotranspiration ($ET_a$), or as potential evapotranspiration from wet surfaces ($ET_p$) with a specific crop type or surfaces covered by large volume of water, such as wetlands or lakes (Stagnitti et al., 1989). $E_{sw}$ is used to represent $ET_p$ from a lake in this paper. $E_{sw}$ from a free water surface has been commonly estimated using the Penman equation (Penman, 1948) that combines the energy budget and mass transfer approaches.

After all, the correct statement in the revised manuscript will read as

'...The deterministic analysis reveals that reference crop evapotranspiration, $ET_o$, set the upper bound for $ET_a$, but the lower bound for $E_{sw}$ in the study area...'

For the sake of clarity, in Objective #1, we look into the relationship not only between reference crop evapotranspiration ($ET_o$) and Actual ET, but also their relations with lake evaporation.

As per the referee's comment, however, we plan to include the following discussion (and additional reference) on the relationships between reference crop evapotranspiration, actual ET, and lake evaporation in the Introduction section:

From a practical standpoint, $ET_p$ has been applied mostly in hydrology, meteorology and climatology; whereas, $ET_o$ has been applied mostly in agronomy, agriculture, irrigation and ecology (Xiang et al., 2020). In particular, $ET_p$ rather than $ET_a$ is a common input for hydrological models, such as HYDRUS, SWAP, SWAT, and MODFLOW-2000 (Li et al., 2016). In drought characterization, $ET_p$, approximated by the $ET_o$, has been used to calculate the aridity index (Kingston et al., 2009, Greve et al., 2019). Although Kristensen and Jensen (1975) reported that $ET_p$ may not be the upper limit of $ET_a$ for all crops or development stages, typically $ET_p$ sets the upper bound for $ET_a$ due to limited water availability for evapotranspiration (Lascano and Bavel 2007, Li et al., 2016). When $ET_a < ET_p$, moisture becomes limited, the air becomes drier and the excess energy heats up the atmosphere, which subsequently increases $ET_p$ (Wang and Zlotkonik, 2012). However, $ET_p \cong ET_a \cong E_{sw}$ holds for wet surface evaporation (Mortan, 1965, Milly and Dunne, 2016). $(ET_a/ET_p)$ represents the

evaporative stress index, (ESI), in which $ET_p$ was approximated by Liu et al. (2019) using the PME-computed PME. The ESI was used to study short term droughts (Choi et al., 2013) and evaluate the irrigation need for crop growth and land classification (Yao, 1974) and water stress using remotely sensed hydrological and ecological properties (Anderson, 2016). If soil moisture data are available, $ET_a$ can be computed by multiplying $ET_p$ by the soil moisture extraction function, defined as the ratio of the measured soil moisture to the field capacity (Lingling et al., 2013). For more comprehensive discussion on different evapostranspiration measures, the readers may refer to the paper by McMahon et al. (2013).

Applications discussed above show that different, yet interrelated, evapotranspiration measures have been used in practice, although they may be converted from one into the other using empirical relations and/or additional hydroclimatic variables. Additional complexities in evapotranspiration calculations and projection are introduced by changes in climate (Milly and Dunne 2016) and land use (Ozdogan and Salvucci, 2004), which alter land surface and lower atmosphere energy budget, and hence, evapotranspiration rates. For example, expansion of irrigated areas in the southern parts of Turkey resulted in $\sim 50\%$ reduction in $ET_p$ and $E_p$ in 23 years due to decreases in wind speed and increases in humidity. Similarly, $ET_o$ exhibited a decreasing trend with an average value of 3 mm/year in the northwest China over 50 years due to decreasing wind speed and radiation and increasing humidity and temperature (Huo et al., 2013).

*1.2 'literature survey is too cursory'*

In the original manuscript, we provided extensive literature review with 50+ references cited in the Introduction and Methods sections in describing different evaporation measures ($ET_o$, $ET_a$, $E_{sw}$), related measurement and calculation techniques, the rationale for their inclusion in our analysis, previously used ML techniques, and the main advantages of our ML model for evapotranspration predictions.

As per the referee's comment, however, we have included additional references (previously reported relations in the literature among the evapotranspiration measures) in the revised manuscript to further improve the discussion and clarity (please see our response 1.1 above for further details).

*1.3 'moving some materials from Section 2 to the Introduction'*

Agree. As per the referee's comment, the Introduction section and Section 2 will be merged, restructured, and revised to enhance the clarity. The discussion on the Edwards aquifer system (between line # 24-30 in the Intro section of the original manuscript) will move to Section 2.1, which will be named 'Study Area and Data Availability' in the revised manuscript.

*1.4 'why existing results are insufficient; why a sophisticated ML to revisit the problem'*

To our knowledge, this is the first ML model proposed & tested to *simultaneously* predict $ET_o$ (while avoiding computationally involved net solar radiation calculations), $E_{sw}$ (while suppressing uncertainties associated with pan evaporation coefficient, measurements, and upscaling), and $ET_a$ (while offsetting high capital & maintenance costs of Eddy Covariance towers) using the standard sets of climatic data obtained from local weather stations, with the exception that water temperature is also required in $E_{sw}$ calculations. From a practical standpoint, it is a cost-effective and efficient computational method that can be used to predict different evaporation measures *simultaneously with high accuracy*, explicitly addressing prediction *'uncertainties'*, different from traditional ML models.

To further provide insights into why we want to use a more sophisticated ML model, we plan to include the following 'research question' in the Introduction section.

Considering the presence of different models for representing evapotranspiration processes as discussed above, we raise the following research question: Can we have a computationally-efficient and unified data-driven machine learning (ML) model to (i) avoid calibration parameters and empirical relations, (ii) calculate different evapotranspiration measures using the standard hydroclimatic data sets, (iii) analyze and report the order of importance of hydroclimatic variables, *while explicitly accommodating their interactions with each other* to identify the most crucial data that need to be acquired for particular evapotranspiration process, (iv) seek new knowledge that may not be readily available from non-probabilistic ML, numerical, or empirical

models, and (v) perform probabilistic predictions over the entire solution space for more accurate assessment of uncertainties related to hydrological predictions?

Moreover, in reference to other ML algorithms listed in the Intro section of the original manuscript, the rationale for the use of a sophisticated ML (hybrid XGBoost-NGBoost) was already provided in line # 51 and 159-165 of the original manuscript. The new ML model is capable of taking 'uncertainties' into consideration. As emphasized in the Intro section, this challenge was brought up by Tang et al. (2018). As illustrated in Fig. 1 and discussed throughout the manuscript, the proposed hybrid ML model is capable of producing not only point predictions (as can be done by traditional ML models), but also a *probability distribution* over the entire solution space (new modeling capability by our hybrid ML model) required for quantifying the *uncertainties* related to hydrological predictions.

More precisely, the hybrid model can produce prediction intervals in addition to point predictions that can effectively inform the user about the model's confidence by quantifying the total number of target datapoints that fall within the specified bounds (e.g. 95%) of the prediction interval. The statistical metrics calculated on the point predictions - such as $R^2$ and RMSE - may not be conclusive about the predictive uncertainties of the resulting models. For example, the statistical metrics for the $ET_a$ model (in Table 1) suggest that the model performs reasonably or fairly, whereas the ML-evaluated 'probabilistic distribution' reveals that 91% of the predicted values are within the 95% prediction interval of the target variable, which provides additional confidence in the model's performance from a practical point-of-view.

*1.5 'motivation needs to be elaborated further in the Intro'*

Agree. Please see our response to Major comment 2 (our reponse 2.1) below.
* * *
**Major comment 2:** Similarly, in Section 2 the motivation of using ML is not clear. What regression methods have been used before? Why existing methods are insufficient in terms of model performance? The authors need to touch on these aspects. Otherwise, the work seems to focus on a new ML algorithm without justification and no baseline results (e.g., multivariate linear regression) were provided.

**Response:**

*2.1 'motivation'*

Although the main motivation for the use of ML was hinted in the Abstract (line # 8-13), throughout the manuscript (e.g., line # 283-287, 300-302, 309-311, 320-322), and in the Conclusion section (line #400 − 412) of the original manuscript, as per the referee's comment, we plan to include the following info in the Intro section of the revised manuscript to enhance the clarity:

The main motivation for development of such a model is to (i) seek an alternative method to the Penman-Monteith equation to avoid computationally intensive net solar radiation computations in $ET_o$ calculations; (ii) overcome uncertainties associated with the pan coefficients, pan evaporation measurements, and upscaling methods for $E_{sw}$ estimates; (iii) offset high capital & maintenance costs of EC towers used for $ET_a$ measurements, and (iv) assessing uncertainties associated with the ML predictions.

*2.2 What regression methods have been used before?*

As discussed in line # 45 - 57 of the original manuscript, previous regression-based methods that were utilized include neural networks, clustering, tree-based ensembles, fuzzy models, multivariate adaptive regression splines, and extreme learning machines. However, the crux of the problem is that none of these ML models are designed to produce prediction intervals for continuous target variables such as $ET_o$, $E_{sw}$, and $ET_a$, which is critical to account for the inherent uncertainties in predictions. Therefore, we applied NGBoost to solve this problem, which is designed to produce prediction intervals for continuous target data. However, as mentioned by Duan et al. [1], "NGBoost is not specifically designed for point estimation" and "better tree-based base learners and regularization (such as XGBoost by Chen and Guestrin [2] ) are more likely to improve performance". These future work indications in their paper led us to develop the hybrid NGBoost-XGBoost model that provides the best of both models in terms of their ability to accurately generate both the prediction intervals and point predictions, respectively (this information and references will also be included in the revised manuscript).

*2.3. 'why existing methods are insufficient in terms of model performance?'*

As elaborated in the original manuscript (e.g., line # 159-165), the existing ML models provide only the point predictions, **but not the probability distributions** over the entire outcome space of continuous target variables. The latter, however, is critical for enhanced uncertainty assessments and building confidence in model predictions.

For example, unlike the other ML methods listed in line #51 of the original manuscript, 'uncertainties' in predictions are accommodated by the NGBoost-XGBoost model, which provided the confidence that at least 90% of the predicted PET, actual ET, and lake evaporation in our ML-based calculations were within 95% of the prediction interval of the target variables. Please see our responses in 1.4 and 2.2 for additional discussion.

*2.4 'without justification and no baseline results?'*

'justification'

For justification for the use of the XGBoost-NGBoost model, pleased see (i) through (v) under 'research question' in our responses 1.4, and (i) through (iv) under 'motivation' in our response 2.1 from the application standpoint, in addition to 'built-in uncertainty calculation' for the predicted target variables from the ML modeling standpoint. This will be highlighted in the Intro section of the revised manuscript to enhance the clarity.

'baseline results'

As per the referee's comment, we have compared the performance of the hybrid model with respect to a baseline linear regression model and included the results in Table 1 below (new analyses and results are marked in blue), which will be included the revised manuscript. The comparison revealed that our proposed hybrid model performs better than the baseline in terms of the statistical metrics for point predictions. But, more importantly, it provides uncertainty estimates through 'probabilistic predictions' (as we further elaborated in our response to Comment # 1.4), which is imperative to practically deploy such models with confidence.

Table 1: Hybrid NGBoost-XGBoost ML model accuracy test with statistical measures and comparison with a baseline linear regression model.

| | Model | Data | RMSE*(mm) | MAE†(mm) | $R^{2\ddagger}$ | $C_f^{\S}$ (%) |
|---|---|---|---|---|---|---|
| $ET_o$ | Baseline | Training data only | 0.205 | 1.364 | 0.984 | - |
| | | Testing data only | 0.191 | 1.374 | 0.986 | - |
| | Hybrid | Training data only | 0.099 | 0.074 | 0.996 | 100 |
| | | Testing data only | 0.139 | 0.102 | 0.992 | 99.4 |
| $E_{sw}$ | Baseline | Training data only | 0.953 | 1.493 | 0.711 | - |
| | | Testing data only | 1.015 | 1.504 | 0.695 | - |
| | Hybrid | Training data only | 0.703 | 0.545 | 0.843 | 99.1 |
| | | Testing data only | 0.918 | 0.736 | 0.750 | 89.9 |
| $ET_a$ | Baseline | Training data only | 0.647 | 1.003 | 0.698 | - |
| | | Testing data only | 0.719 | 1.011 | 0.643 | - |
| | Hybrid | Training data only | 0.388 | 0.291 | 0.891 | 99.9 |
| | | Testing data only | 0.533 | 0.411 | 0.804 | 91 |

(*) Root mean square error; † Mean absolute error; ‡ Correlation Coefficient; § Percentage of datapoint within the model's 95% prediction interval.

**Major comment 3:** The ML pipeline is not clear. A diagram is needed to show inputs and output to the ML model. Around L185, the authors simply spelled out the inputs, without much reasoning. Why these features are selected? What is the lead time of prediction? The promise of ML is not so much for well gauged sites, but for sites with a lot of missing data.

**Response:**

*3.1 'diagram'*

As per the referee's comment, we have revised the flow chart in Fig. 1 of the original manuscript, as shown below, which we plan to include in the revised manuscript.

[Figure]

Figure 1: Conceptual representation of the hybrid NGBoost-XGBoost model for $ET_o$, $E_{sw}$, and $ET_a$ prediction.

*3.2 'why these features are selected?'*

As per the referee's comment, we will include this information in the manuscript. Briefly,

The climatic variables ($T_a$, $P$, $RH$, $u_2$, $R_s$) in Eq. 1 (PME) were chosen as the features for the $ET_o$ predictive model. The same climatic variables were used as the features for the $ET_a$ predictive model, in addition to $ET_o$ to quantify its contribution to $ET_a$. $T_{sw}$ in Eqs. 6 and 7 (Meyer's formula) was added as a new feature to the climatic variables in the $E_{sw}$ predictive model. Moreover, 'month' was chosen as an additional feature in all predictive models based on the observed seasonality in $T_{sw}$ data, $ET_a$ measurements from the EC tower, and expected seasonality in soil moisture content at the site where the the EC tower is located.

*3.3 what is the lead time in predictions?*

If the Referee refers to the 'training time' of the $ET_o$, $E_{sw}$, and $ET_a$ predictive models by the 'lead time in predictions', this information was already included in the 'Predictive ML Models' section of the original manuscript. Briefly, the total training time of the $ET_o$, $E_{sw}$, and $ET_a$ predictive models are $\sim$ 30 min, $\sim$ 6 min, and $\sim$ 9 min. As mentioned in the manuscript, the models were developed using an Intel Core i9-9980XE processor and 64 GB RAM computer. Otherwise, please provide clarification on what the Referee means by the 'lead time in predictions'.

*3.4 'The promise of ML is not so much for well gauged sites, but for sites with a lot of missing data.'*

There is a growing number of studies in the literature, in which ML models have been primarily used for assessment, simulation, estimation, solving and capturing nonlinear complexity, and projection of various hydrologic processes, including for example, precipitation, evapotranspiration, droughts, floods, groundwater levels ([3]–[15] are just a few examples from a long list of recent articles), different from other types of studies,

in which the ML methods are primarily used for imputing missing data, as the Referee mentioned about. These two types applications of the ML models (simulation & prediction vs. data imputation) were emphasized in Refs. [7] and [16], as an example, without favoring one over another, as the ML methods have emerged as promising modeling tools in both application fields, but not as a promising tool only for 'ungauged' systems with lots of missing data, as the Referee suggested.

In the first type of applications discussed above (which is well-aligned with the the main scope of our current manuscript), data imputation are often taken care of outside the ML modeling (e.g., by removing missing or suspicious data as in Refs. [17], [18] or using statistical packages as in Ref. [3]). Thus, the main goal of the ML modeling in this case is to decipher the nonlinear dynamics between the input (predictive variables) and output (target variables) and use such information for prediction & projection of target variables without requiring detailed physical information (e.g., constitutive equations reflecting physical laws) on the investigated system, which would otherwise require a large volume of data, non-unique calibration processes, and high computational costs, as emphasized by [7] and [10], as an example.

After all, we are well aware that ML modeling for hydrological simulations and predictions and ML modeling for data imputations are two different (but, sometimes integrated) applications. We have another manuscript currently in review at one of the Artificial Intelligence journals, focusing on a new ML model -based on the transfer learning approach- to impute long-stretches (a few months to a year) of missing data. This is another application of the ML model the referee is referring to. In brief, in light of current literature and our experience, we believe ML modeling is as important to analyze, predict, and project nonlinear dynamics between predictor and target variables as to impute missing data. And, our current manuscript focused on the former application.
* * *
**Major comment 4:** Variable importance calculation is well established for tree-based method, which entails finding whether a variable is selected to split on during the tree building process, and how much the squared error (over all tress) us improved or reduced as a result. Why a new variable importance method is needed?

**Response:** The typical variable importance calculations are equivalent to a sensitivity analysis, measuring *relative* contribution of a specific predictor variable to the target observation *without* accommodating the dynamic interaction of that specific predictor variables with the other predictor variables. In contrast, as mentioned in the original manuscript, the Shapley value is the average marginal contribution of each feature value across all possible combinations of features. Thus enabling (i) global interpretability: the collective SHAP values can show how much each feature contributes to the target, which is similar to the traditional tree-based permuted feature importance, however, the SHAP plots can additionally explain the positive or negative relationship between each feature value and the target (see Fig. 10 in the manuscript); (ii) local interpretability: traditional variable importance plots only show the results across the entire dataset, but not on each individual datapoints. In contrast, with the new SHAP-based technique each observation gets its own set of SHAP values (see Fig. 12 in the manuscript). This greatly increases the transparency of the ML models and reveals new insights. For example, Fig. 12 reveals new knowledge on low $ET_a$ predictions despite high $ET_o$ & low RH measures in hot and dry summer, which will be critical in future climate scenarios in Texas or elsewhere. Such insights are not available from the traditional tree-based variable importance plots.
* * *
**Minor comment:** In Abstract, the authors concluded "the deterministic analysis reveals that ETo set the upper bound for Eta", isn't this expected?

**Response**
As we discussed in our response to the Major Comment #1, $ET_o$ represents the reference crop evapotranspiration, not the potential evapotransporation (this was a typo in the original manuscript and we corrected in the revised version).

Fig. 8b show that $ET_a < ET_o$ for most of the time, except for October through December of 2019, in which $ET_a \cong ET_o$, which we did not expect initially for a semi-arid region in Texas.

Having said this, $ET_a < ET_o$ confirms the reliability and confidence in measured local climatic data from the local weather stations, our $ET_o$ calculations using the PME, and the actual ET measurements from the EC tower. The validity of this relation confirms that the ML models were operated on 'physically reasonable' input data sets, eliminating concerns on potential uncertainties or inconsistencies in the input data used in ML analysis.

**References**

[1]   T. Duan, A. Avati, D. Y. Ding, S. Basu, A. Y. Ng, and A. Schuler, "Ngboost: Natural gradient boosting for probabilistic prediction", *ArXiv preprint arXiv:1910.03225*, 2019.

[2]   T. Chen and C. Guestrin, "Xgboost: A scalable tree boosting system", in *Proceedings of the 22nd ACM SIGKDD International Conference on Knowledge Discovery and Data Mining*, 2016, pp. 785–794.

[3]   P. Malakar, A. Mukherjee, S. Bhanja, D. Saha, R. K. Ray, S. Sarkar, and A. Zahid, "Importance of spatial and depth-dependent drivers in groundwater level modeling through machine learning", *Hydrol. Earth Syst. Sci. Discuss.*, 2020, in review. DOI: `https://doi.org/10.5194/hess-2020-208`.

[4]   M. K. Nema, D. Khare, and S. K. Chandniha, "Application of artificial intelligence to estimate the reference evapotranspiration in sub-humid doon valley", *App. Water Sci.*, vol. 7, no. 7, pp. 3903–3910, 2017.

[5]   S. Pan, N. Pan, H. Tian, P. Friedlingstein, S. Sitch, H. Shi, V. Arora, V. Haverd, A. Jain, E. Kato, S. Lienert, D. Lombardozzi, J. Nabel, C. Ottlé, B. Poulter, S. Zaehle, and S. Running, "Evaluation of global terrestrial evapotranspiration using state-of-the-art approaches in remote sensing, machine learning and land surface modeling", *Hydrol. Earth Syst. Sci.*, vol. 24, pp. 1485–1509, 2020.

[6]   C. Chong, H. Wei, Z. Han, X. Yaru, and Z. Mingda, "A comparative study among machine learning and numerical models for simulating groundwater dynamics in the Heihe River Basin, northwestern China", *Sci. Rep.*, vol. 10, p. 3904, 2020.

[7]   R. Taormina, K.-W. Chau, and R. Sethi, "Artificial neural network simulation of hourly groundwater levels in a coastal aquifer system of the Venice lagoon", *Eng. Appl. Artif. Intell.*, vol. 25, no. 8, pp. 1670–1676, 2012.

[8]   A. Zhang, J. Winterle, and C. Yang, "Performance comparison of physical process-based and data-driven models: A case study on the Edwards Aquifer, USA", *Hydrogeol. J.*, 2020.

[9]   Z. A. Al-Sudani, S. Q. Salih, A. Sharafati, and Z. M. Yaseen, "Development of multivariate adaptive regression spline integrated with differential evolution model for streamflow simulation", *J. Hydrol.*, vol. 573, pp. 1–12, 2019.

[10]   T. Rajaeea, H. Ebrahimia, and V. Nouranib, "A review of the artificial intelligence methods in groundwater level modeling", *J. Hydrol.*, vol. 572, pp. 336–351, 2019.

[11]   S. Jovic, B. Nedeljkovic, Z. Golubovic, and N. Kostic, "Evolutionary algorithm for reference evapotranspiration analysis", *Comput. Electron. Agric.*, vol. 150, pp. 1–4, 2018.

[12]   L. Schmidt, F. Heße, S. Attinger, and R. Kumar, "Challenges in applying machine learning models for hydrological inference: A case study for flooding events across Germany", *Water Resour. Res.*, vol. 56, e2019WR025924, 2020.

[13]   O. Kisi and M. Alizamir, "Modelling reference evapotranspiration using a new wavelet conjunction heuristic method: Wavelet extreme learning machine vs wavelet neural networks", *Agr. Forest Meteorol.*, vol. 263, pp. 41–48, 2018.

[14]   N. Khan, D. A. Sachindra, S. Shahid, K. Ahmed, M. S. Shiru, and N. Nawaz, "Prediction of droughts over Pakistan using machine learning algorithms", *Adv. Water Resour.*, vol. 139, p. 103 562, 2020.

[15] K. Ahmed, D. A. Sachindra, S. Shahid, Z. Iqbala, N. Nawaz, and K. Najeebullah, "Multi-model ensemble predictions of precipitation and temperature using machine learning algorithms", *Atmos. Res.*, vol. 236, p. 104 806, 2020.

[16] X. Dou and Y. Yang, "Evapotranspiration estimation using four different machine learning approaches in different terrestrial ecosystems", *Comput. Electron. Agric.*, vol. 148, pp. 95–106, 2018.

[17] J. Fan, W. Yue, L. Wu, F. Zhang, H. Cai, X. Wang, X. Lu, and Y. Xiang, "Evaluation of SVM, ELM and four tree-based ensemble models for predicting daily reference evapotranspiration using limited meteorological data in different climates of China", *Agr. Forest Meteorol.*, vol. 263, pp. 225–241, 2018.

[18] X. Lu, Y. Ju, L. Wu, J. Fan, F. Zhang, and Z. Li, "Daily pan evaporation modeling from local and cross-station data using three tree-based machine learning models", *J. Hydrol.*, vol. 566, pp. 668–684, 2018.

---

## Referee Comment (RC2) · Anonymous Referee #2 · 2 Oct 2020

This paper is not well motivated and lacks focus. The point of E0 modeling by XGBoost was to saved computation. The P-M equation is very simple and the computational time is negligible. Ea and Esw might have some value, but there lacked comparisons with other methods. Why can't you estimate a relationship between E0 and Ea? Instead of XGBoost, why can't we use linear regression or autoregressive function? What is it in here that we cannot get elsewhere?

The author then introduced Shapley values and claimed it a game-theory-based feature importance ranking. If the purpose was to introduce a new feature ranking scheme, it should be compared to other methods to show its validity. This dilutes the focus and

looks like a demonstration of software capabilities.

The organization and writing of the paper are poor. There needs to be a clear positioning of the paper in the context of what has been known. The introduction is supposed to position the work in the context of the literature. For ET, that is a lot of literature. Although the authors cited some ML for ET papers, they were discussed very surficially. Instead the introduction was extremely surficial and entirely omitted all the literature in ET in hydrology. Then the authors spent 8 figures presenting stuff that has no modeling components. It was very hard to understand what were the inputs and what were the outputs from Methods. This was instead discussed in the Results section. The authors used "unprecedented" and "for the first time" very casually. There are things we do and things we don't do and there must be a clear reason why we do them.

---

## Author Comment (AC2) · 5 Oct 2020

We have interspersed our responses between the questions and comments. New texts, figures, Tables planned to be included in the manuscript in response to the comments are marked in blue in our responses below.
* * *
**Referee** $\#2$:
* * *
**Comment 1.** This paper is not well motivated and lacks focus.

**Response:**

*1.1 'not well motivated'*

Without any specific reasoning, it is unclear to us why the Referee considered our work to be 'not well-motivated'.

However, as already articulated in the Abstract, Introduction, main body (e.g., line $\#$ 282-287 on page 13), and in the Conclusions section of the original manuscript, the main motivations for the work and development of a new probabilistic Machine Learning (ML) model are to

(i) seek an alternative method to the Penman-Monteith equation to avoid computationally involved net solar radiation computations in $ET_o$ calculations,

(ii) overcome uncertainties associated with the pan coefficients and pan evaporation measurements,

(iii) offset high capital & maintenance costs of EC towers used for $ET_a$ measurements, and

(iv) assess uncertainties associated with the ML predictions

The first three motivation and the associated findings have been articulated throughout the manuscript (e.g., the last paragraph on page 13). Motivation (iv) was associated with the challenge brought up by Tang et al. (2018), which was already cited and discussed in line $\#$ 52 of the original manuscript.

Although we believe that the main motivation of the work was clear in the original manuscript, we plan to include a stand-alone 'motivation' discussion, as discussed above, in the Introduction section of the revised manuscript to enhance the clarity.

*1.2 'lacks focus'.*

Again, without any specific reasoning, it is unclear to us why the Referee considered the manuscript 'lacks focus'.

However, as already highlighted in the Title, noted in the Abstract, and discussed throughout the manuscript, the manuscript **focuses** on the use of a novel probabilistic ML model (a hybrid XGBoost-NGBoost model) to calculate the daily and monthly $ET_o$, $E_{sw}$, and $ET_a$ using local hydroclimatic data, and to identify & quantify the importance and interactions of features (hydroclimatic variables) in predicting targeted evapotranspiration measures using Shapley values - a method from the coalition game theory.

This information was already provided in **'Objectives** $1-3$**'** in line $\#$ $39-44$ (on page 2) of the Introduction section of the manuscript, which was the **main focus** of the subject manuscript and associated with the main conclusions elaborated in line $\#$ $410-428$ of the Conclusion sections (on pages $21-22$) of the manuscript. This is also consistent with the highlights we provided in line $\#$ $4-14$ of the Abstract.

Briefly, the main focus of the manuscript was consistently articulated in the Abstract, Introduction, throughout the text of the manuscript, and in the Conclusion section. So, we believe the main focus of the work has

been carried out consistently throughout the manuscript.

As we highlighted in the original manuscript, the use of the hybrid XGBoost-NGBoost in $ET_o$, $E_{sw}$, and $ET_a$ prediction, and the use of Shapley values (computed based on the coalition game theory) to determine the importance (considering their interactions explicitly) of the features (i.e., hydroclimatic variables) on the targeted evapotranspiration are unprecedented. Although we discussed the rationale and motivation for the use of these new methods in evapotranspiration prediction in the manuscript, we provided further justification for the use of these novel modeling features in our responses to the Referee's comments below.
* * *
**Comment 2.** The point of E0 modeling by XGBoost was to saved computation. The P-M equation is very simple and the computational time is negligible.

**Response:**

The main motivation of the $ET_o$ modeling using the hybrid XGBoost-NGBoost model was to reduce (or eliminate) the number of equations (and associated parameters) implemented with the time-series of climate or derived data in $ET_o$ calculations. Eq. (1) may look simple, yet it is computationally involved (see Fig. 1 for the main calculation steps to generate Fig. 5d in the manuscript). Its full-scale implementation is not 'very simple' as the Referee suggested, but requires very diligent coding, critical implementation steps (e.g., sunset time in calculating extra terrestrial solar radiation, comparison of cloudiness computed by the PME to the measured cloudiness) in calculations, and thorough analysis.

Using the hybrid XGBoost-NGBoost model in the testing phase of the ML modeling, all the equations listed below (in reference to equation numbers in the FAO report by Allen et al. (1998, cited in the manuscript)) and their calculations in Fig. 1 below were replaced by 'Fig. 9a' in the manuscript (i.e., considerable reductions in computational steps). On the other hand, 'Fig. 9a' in the manuscript was generated using only daily $R_s$, $T_a$, $RH$, $P$, and $u_2$ without any other intermediate calculations and net solar radiation calculations. The equations eliminated using the ML model to calculate $ET_o$ involved:

Hourly mean saturation vapor pressure (Eq.11, p.36, main document, FAO56, Allen et al., 1998)
Hourly mean actual vapor pressure (Eq.54, p.74, main document, FAO56, Allen et al., 1998)
Slope of the saturation pressure curve (Eq.13, p.37, main document, FAO56, Allen et al., 1998)
Psychrometric constant (Eq.8, p.32, main document, FAO56, Allen et al., 1998)
Longwave outgoing radiation (Eq.39, p.52, Box 9, FAO56, Allen et al., 1998)
Solar time angle midpoint of hourly period (Eq.31, p.48, Box 9, FAO56, Allen et al., 1998)
Seasonal correction for solar time (Eqs.32-33, p.48, Box 9, FAO56, Allen et al., 1998)
Inverse relative distance Earth-Sun (Eq. 23, p.46, Box 9, FAO56, Allen et al., 1998)
Solar declination (Eq. 24, p.46, Box 9, FAO56, Allen et al., 1998)
Sunset hour angle (Eq. 25, p.46, Box 9, FAO56, Allen et al., 1998)
Extra-terrestrial radiation for hourly period (Eq. 28, p.47, main document, FAO56, Allen et al., 1998)
Relative solar radiation (cloudiness) (Eq. 3-15, p.226, main document, FAO56, Allen et al., 1998)
Net radiation (Eq. 40, p.53 main document, FAO56, Allen et al., 1998)

In brief, the hybrid XGBoost-NGBoost model effectively and significantly reduced the computational steps in $ET_o$ while eliminating computationally-involved net solar radiation calculations, including also extra-terrestrial radiation and longwave radiation calculations.

[Figure]

Figure 1: The main steps in $ET_o$ calculations using the PME.

**Comment 3.** Ea and Esw might have some value, but there lacked comparisons with other methods.

**Response:**

It is unclear what the Referee meant by 'other methods' in the comment above, as no specifics have been provided for 'other methods'.

However, daily and monthly $E_{sw}$ 'measurements'(upscaled from pan evaporation data) were already compared against the computed $E_{sw}$ via Meyer's formula ('another method', if we adopt the Referee's terminology), using Eqs. 6 and 7 in the manuscript in Fig. 6 of the manuscript, and against $ET_o$ (using Eq. 1) in Fig. 7 of the manuscript. As already mentioned in line #149 of the manuscript, Eq. 6 is the best form of Meyer Formula to predict daily $E_{sw}$ from free water surface − the associated references were already provided just before Eq. 6 in the original manuscript.

It is also unclear from the comment above why we need another model to predict $E_a$, as the daily $E_a$ (and hence, monthly aggregated $E_a$) in Fig. 8 are the 'measured data' acquired from the EC tower. We used these data to test the predictive performance of our new hybrid XGBoost-NGBoost model.

As for the use of probabilistic ML model for $E_{sw}$ and $ET_a$ prediction, as discussed in the Abstract, in the main body of the manuscript, and in the Conclusion section, the use of a hybrid XGBoost-NGBoost model would eliminate high capital and maintenance costs of the EC towers (as mentioned in line #276 of the manuscript, the capital cost for the EC tower was $40,000 and required frequent maintenance) and overcome uncertainties associated with pan evaporation and pan coefficient in $E_{sw}$, which are critical issues in practice.

**Comment 4.** Why can't you estimate a relationship between E0 and Ea? Instead of XGBoost, why can't we use linear regression or autoregressive function?

**Response:** Fig.11c of the manuscript reveals that the statistical correlation between the daily $ET_o$ and daily $ET_a$ is not high ($R^2 = 0.74$). The underlying reasoning was already elaborated in line $\#369-391$ (on page 19) of the original manuscript. Moreover, the relation between $ET_o$ and $ET_a$ is clearly nonlinear in Fig. 8. Therefore, the linear regression method that the Referee suggested is not expected to perform well here. To demonstrate this, we set another ML model based on Linear Regression model (labeled as 'Baseline') and compared its performance against the hybrid NGBoost-XGBoost model (labeled as 'Hybrid') in predicting $ET_o$, $E_{sw}$, and $ET_a$ in Table 1.

As expected, Table 1 below shows that the hybrid NGBoost-XGBoost model clearly and consistently outperformed the baseline linear regression model in predicting $ET_o$, $E_{sw}$, and $ET_a$ in terms of the 'statistical metrics' for point predictions (e.g., in terms of RMSE, MSE, $R^2$). But, more importantly, the hybrid NGBoost-XGBoost model provided uncertainty estimates through 'probabilistic predictions' (unlike the Linear Regression model and/or other previously developed ML models discussed in line $\# 45-57$ (on page 2) of the original manuscript), which is imperative to practically deploy such models with confidence.

As for the Referee's suggestion on autoregressive models, autoregressive models are beyond the scope of this work, because (i) the prediction of $ET_o$, $E_{sw}$, and $ET_a$ is a 'multivariate problem' (please see Eqs. 1, 6, and 7) and autoregressive models cannot unveil the relative importance and interactions of each features (hydroclimatic variables) on the targeted evapotranspiration measure as in Figs. 10 and 12, which are critical in practice, as evident from a number of studies we reviewed, discussed, and compared against our findings on pages $16-19$ of the manuscript; (ii) the presence of lagged variables (t-1, t-2, etc.) in autoregressive models leads to accumulation of errors, which is unsuitable for long-term predictions, and also, the lagged variables take away the focus from the climatic variables, due to their high correlation with the target variable, thereby not allowing us to correctly identify and detect the importance and interactions between the hydroclimatic variables using Shapley values; and (iii) the standalone autoregressive models cannot produce probabilistic predictions, which were made possible with the proposed hybrid model as in Fig. 9, which is imperative for uncertainty analysis, and hence, for the use of such models with confidence in practice.

Table 1: Hybrid NGBoost-XGBoost ML model accuracy test with statistical measures and comparison with a baseline linear regression model.

|  | Model | Data | RMSE*(mm) | MAE†(mm) | $R^2$ ‡ | $C_f^\S$ (%) |
|---|---|---|---|---|---|---|
| $ET_o$ | Baseline | Training data only | 0.205 | 1.364 | 0.984 | - |
|  |  | Testing data only | 0.191 | 1.374 | 0.986 | - |
|  | Hybrid | Training data only | 0.099 | 0.074 | 0.996 | 100 |
|  |  | Testing data only | 0.139 | 0.102 | 0.992 | 99.4 |
| $E_{sw}$ | Baseline | Training data only | 0.953 | 1.493 | 0.711 | - |
|  |  | Testing data only | 1.015 | 1.504 | 0.695 | - |
|  | Hybrid | Training data only | 0.703 | 0.545 | 0.843 | 99.1 |
|  |  | Testing data only | 0.918 | 0.736 | 0.750 | 89.9 |
| $ET_a$ | Baseline | Training data only | 0.647 | 1.003 | 0.698 | - |
|  |  | Testing data only | 0.719 | 1.011 | 0.643 | - |
|  | Hybrid | Training data only | 0.388 | 0.291 | 0.891 | 99.9 |
|  |  | Testing data only | 0.533 | 0.411 | 0.804 | 91 |

(*) Root mean square error; † Mean absolute error; ‡ Correlation Coefficient; § Percentage of datapoint within the model's 95% prediction interval.

As per this comment, Table 1 will be included in the revised manuscript to demonstrate that the hybrid

NGBoost-XGBoost model outperformed the traditional Linear Regression models, which was suggested by the Referee, in analyzing nonlinear relations between $ET_o$ and $ET_a$.
* * *
**Comment 5.** What is it in here that we cannot get elsewhere?

**Response:**

As elaborated in the original manuscript (e.g., line $\#\,45-57$ and $159-165$) and in our response to the previous comment, the existing ML models provide only the point predictions, **but not the probability distributions** over the entire outcome space of continuous target variables. The latter, however, is critical for enhanced 'uncertainty' assessments and building confidence in model predictions of $ET_o$, $ET_a$, and $E_{sw}$ in practice.

For example, unlike the other ML methods listed in line $\#51$ of the original manuscript, 'uncertainties' in predictions are accommodated by the hybrid NGBoost-XGBoost model, which provided the confidence that at least $90\%$ of the predicted $ET_o$, $ET_a$, and $E_{sw}$ in our ML-based calculations were within $95\%$ of the prediction interval of the target variables. This conclusion **cannot be obtained** from the previously developed ML model, or from the Linear Regression and Autoregressive models that are suggested by the Referee.

Moreover, typical features (independent variables) importance calculations are equivalent to a sensitivity analysis, measuring *relative* contribution of a specific predictor variable to the target observation, but *without* accommodating the dynamic interaction of that specific predictor variable with the other predictor variables. In contrast, as mentioned in the original manuscript, the Shapley value is the average marginal contribution of each feature value across all possible combinations of features. Using the Shapley values, Fig. 12 reveals new knowledge on low $ET_a$ predictions despite high $ET_o$ & low RH measures in hot and dry summer, which will be critical in future climate scenarios in Texas or elsewhere. Such insights and new information **cannot be** obtained from the traditional features importance analysis and plots.
* * *
**Comment 6.** The author then introduced Shapley values and claimed it a game-theory-based feature importance ranking. If the purpose was to introduce a new feature ranking scheme, it should be compared to other methods to show its validity? This dilutes the focus and looks like a demonstration of software capabilities.

**Response:**

*6.1 '...and claimed it a game-theory-based '*

We did not 'claim' that Shapley values are a game theory based importance ranking. It is a fact. Please refer to the excellent study by Lundberg et al. (2020; Nature Machine Intelligence), which was already cited in the original manuscript.

*6.2 'it should be compared to other methods to show its validity '*

Again, it is unclear which other methods the Referee is referring to in this comment. Without any specifics in this comment, here we assume that the Referee might be referring to tree-based features importance analysis or sensitivity or correlation based feature importance analysis.

To the best of our knowledge, although the Shapley values have not been implemented in hydrological problems in the literature to date, its importance has already been highlighted in the recent literature. For example, Schmidth et al. (2020, Water Resources Research, 10.1029/2019WR025924) noted that :

'*Therefore, other model-agnostic methods that have been made available in the recent past like* **Shapley values** *or Local Interpretable Model-agnostic Explanations should be applied additionally to* **rule out any bias** *that is specific to the respective method.*'

Moreover, as already elaborated in the original manuscript, the typical variable importance calculations are equivalent to a sensitivity analysis, measuring *relative* contribution of a specific predictor variable to the target

observation *without* accommodating the dynamic interaction of that specific predictor variable with the other predictor variables. In contrast, as mentioned in the original manuscript, the Shapley value is the average marginal contribution of each feature value across all possible combinations of features.

Unlike the tree-based feature importance methods, the Shapley values method enables:

(i) global interpretability: the collective SHAP values can show how much each feature contributes to the target, which is similar to the traditional tree-based permuted feature importance, however, the SHAP plots can additionally explain the positive or negative relationship between each feature value and the target (see Fig. 10 in the manuscript), and

(ii) local interpretability: traditional variable importance plots only show the results across the entire dataset, but not on each individual datapoints. In contrast, with the new SHAP-based technique each observation gets its own set of SHAP values (see Fig. 12 in the manuscript). This greatly increases the transparency of the ML models and reveals new insights.

On the other hand, if the Referee is referring to traditional sensitivity analysis or correlation maps, as already shown in Fig. 11 of the manuscript, such analysis cannot accommodate the dynamic interaction of a specific predictor variable with the other predictor variables in calculating the contribution of a specific predictor variable to the target evaporation measures ($ET_o$, $ET_a$, $E_{sw}$).

In brief, as elaborated in the original manuscript, the Shapley values presented in Fig. 10 captured the underlying physics of the evapotranspiration process reasonably well. Because the tree-based methods, traditional sensitivity, and correlation analysis are incapable of accounting for such dynamic interactions among the predictor variables simultaneously (which is made possible with the cooperative game theory-based Shapley values) in calculating the contribution of a specific predictor variable to the target output, a comparison of the results from the Shapley values to the results from the other traditional methods would be uninformative and inconclusive. For example, neither the sensitivity and correlation analysis nor tree-based feature importance methods is capable of producing the results and capturing the conclusions in Fig. 12, and hence, making such comparison is impossible.

*6.3 'This dilutes the focus and looks like a demonstration of software capabilities'*

With all due respect, we believe that it does not dilute the focus, as the Shapley values-based feature importance analysis constitutes 'Objective 3' (one one the **main focuses**) of the manuscript, as described in line $43-44$ of the manuscript. Moreover, this analysis is important to (i) enhance the transparency of the ML model by identifying & quantifying dynamic interactions between the predictor (hydroclimatic) variables and the target evapotranspiration measures; and (ii) assess the acceptability of applicability of the simplified evapotranspiration models implemented at basins with scarce data (see the discussion in the last paragraph of page 16 of the manuscript). Thus, the Shapley-value based feature importance analysis is an essential component of the hybrid XGBoost-NGBoost modeling, and has nothing to do with demonstration of software capabilities (we even never used the term 'software' in our manuscript, but we focused on a new ML model/method), as the Referee suggested in the comment above.
* * *
**Comment 7.** The organization and writing of the paper are poor.?

**Response:**
We separated this part of the comment, as it refers to the overall 'manuscript', different from the critiques on the Intro section in the following sentence of the comment. However, it is unclear to us what parts of the writing in the overall 'manuscript' (other than the critiques on the Intro section below) are poor and why? Similarly, what parts of the structure of the overall manuscript is poor and why? Without specific reasoning, it is not

possible to respond to this comment.

Having said this, we followed the 'guideline for manuscript preparation' when we prepared (and organized) the manuscript. The Introduction section provides the background information, main objectives, motivation for the study, rationale for the chosen methods, and briefly explains critical findings. Next, we provide more detailed information about the Methods and Available Data used in the ML analysis. These sections are followed by the Results and Discussion, and the main findings and contributions are elaborated in the Conclusion section. It is unclear to us why the Referee considered this flow of information and manuscript organization to be poor.

More importantly, because our manuscript was accepted for the peer review after the initial screening by the Editors, we believe the overall structure and the writing style of the manuscript should be acceptable and in compliance with the Journal's guideline. We feel like such comments are probably not applicable at this phase of the review process, but certainly without specific reasoning, they are not constructive at all.
* * *
**Comment 8.** There needs to be a clear positioning of the paper in the context of what has been known. The introduction is supposed to position the work in the context of the literature.

**Response:**

*8.1 'clear positioning of the paper in the context of what has been known '*

This has been addressed in the original manuscript, but we agree with the comment that it may still require further improvements. In the first two paragraphs below, we elaborate how this has already been addressed in the manuscript, and in the subsequent paragraphs we explain how to implement further improvements.

We noted in line $\#45-57$ on page 2) that various ML models/methods have been recently developed and used for ET prediction **without accounting for inherit uncertainties**. This issue was brought up by Tang et al (2018), as we stated in the Introduction section, and we confronted this challenge in our manuscript for the first time in predicting $ETo$, $ET_a$, $E_{sw}$ prediction (as described in the Introduction section). This laid the foundation for motivation (iv), as further elaborated in our response 1.1 above, and Objective 2 (line $\#$ 2) in the Introduction section of the original manuscript.

Moreover, one of the main objectives is to develop a probabilistic ML model to predict $ET_o$, $ET_a$, and $E_{sw}$ simultaneously from standard hydroclimatic data sets. To our best of knowledge, there is 'no' such a probabilistic ML model in the literature to predict all these evapotranspiration measures; therefore, we used the term 'unprecedented', clearly positioning the paper in the context of what has been known in the literature, as the Referee suggested in the comment above. In brief, this thought process has already been embedded in the manuscript. But, we agree with the Referee's comment that it needs further improvements and we elaborate below how we plan to incorporate such improvements.

As part of further improvements, we plan to include the following discussion and additional references in the Introduction section, as we elaborated in our response to Referee 1's comments, which is also applicable to this comment.

Evapotranspiration can be computed as reference crop evapotranspiration ($ET_o$), actual evapotranspiration ($ET_a$), or as potential evapotranspiration from wet surfaces ($ET_p$) with a specific crop type or surfaces covered by large volume of water, such as wetlands or lakes (Stagnitti et al., 1989). $E_{sw}$ is used to represent $ET_p$ from a lake in this paper. $E_{sw}$ from a free water surface has been commonly estimated using the Penman equation (Penman, 1948) that combines the energy budget and mass transfer approaches.

From a practical standpoint, $ET_p$ has been applied mostly in hydrology, meteorology and climatology; whereas, $ET_o$ has been applied mostly in agronomy, agriculture, irrigation and ecology (Xiang et al., 2020). In particular, $ET_p$ rather than $ET_a$ is a common input for hydrological models, such as HYDRUS, SWAP, SWAT, and MODFLOW-2000 (Li et al., 2016). In drought characterization, $ET_p$, approximated by the $ET_o$, has been used to calculate the aridity index (Kingston et al., 2009, Greve et al., 2019). Although Kristensen and Jensen

(1975) reported that $ET_p$ may not be the upper limit of $ET_a$ for all crops or development stages, typically $ET_p$ sets the upper bound for $ET_a$ due to limited water availability for evapotranspiration (Lascano and Bavel 2007, Li et al., 2016). When $ET_a < ET_p$, moisture becomes limited, the air becomes drier and the excess energy heats up the atmosphere, which subsequently increases $ET_p$ (Wang and Zlotkonik, 2012). However, $ET_p \cong ET_a \cong E_{sw}$ holds for wet surface evaporation (Mortan, 1965, Milly and Dunne, 2016). $(ET_a/ET_p)$ represents the evaporative stress index, (ESI), in which $ET_p$ was approximated by Liu et al. (2019) using the PME-computed PME. The ESI was used to study short term droughts (Choi et al., 2013) and evaluate the irrigation need for crop growth and land classification (Yao, 1974) and water stress using remotely sensed hydrological and ecological properties (Anderson, 2016). If soil moisture data are available, $ET_a$ can be computed by multiplying $ET_p$ by the soil moisture extraction function, defined as the ratio of the measured soil moisture to the field capacity (Lingling et al., 2013). For more comprehensive discussion on different evapostranspiration measures, the readers may refer to the paper by McMahon et al. (2013).

Applications discussed above show that different, yet interrelated, evapotranspiration measures have been used in practice, although they may be converted from one into the other using empirical relations and/or additional hydroclimatic variables. Additional complexities in evapotranspiration calculations and projection are introduced by changes in climate (Milly and Dunne 2016) and land use (Ozdogan and Salvucci, 2004), which alter land surface and lower atmosphere energy budget, and hence, evapotranspiration rates. For example, expansion of irrigated areas in the southern parts of Turkey resulted in $\sim 50\%$ reduction in $ET_p$ and $E_p$ in 23 years due to decreases in wind speed and increases in humidity. Similarly, $ET_o$ exhibited a decreasing trend with an average value of 3 mm/year in the northwest China over 50 years due to decreasing wind speed and radiation and increasing humidity and temperature (Huo et al., 2013).

*8.2 'introduction is supposed to position the work in the context of the literature '*

**Response:**

We plan to include the following statement in the revised Introduction after the extended literature review provided in our response above to further strengthen the position of the work in the context of the literature:

Considering the presence of different models for evapotranspiration processes as discussed above, we raise the following research questions: Can we have a computationally-efficient and unified data-driven machine learning (ML) model to (i) avoid calibration parameters (e.g., pan coefficients) and empirical relations (e.g.,, Meyer formula), (ii) calculate different evapotranspiration measures ($ET_a$, $E_{sw}$, and $E_a$) using the standard hydroclimatic data sets, (iii) analyze and report the order of importance of hydroclimatic variables, *while explicitly accommodating their interactions with each other* to identify the most crucial datasets that need to be acquired for particular evapotranspiration process, (iv) seek new knowledge that may not be readily available from non-probabilistic ML, numerical, or empirical models, and (v) perform probabilistic predictions over the entire solution space for more accurate assessment of uncertainties related to hydrological predictions?

Moreover, a stand alone 'motivation' paragraph, as discussed in our Response 1.1 will also be included in the Introduction section to highlight the main contributions of this work beyond what is already available in the existing literature.
* * *
**Comment 9.** For ET, that is a lot of literature. Although the authors cited some ML for ET papers, they were discussed very surficially. Instead the introduction was extremely surficial and entirely omitted all the literature in ET in hydrology. ?

**Response:** As we discussed in our response above, the additional literature review on the different evapotranspiration measures and their interrelations (as per Referee 1's comment) is planned to be included in the revised manuscript. If the Referee 2 has any additional suggested references relevant to our work, we are open to his/her suggestions and we will address them in the manuscript as appropriate.

However, although some literature on the evapotranspiration measures might be missing in the original

manuscript, we do not agree with the statement in the comment above that 'the authors entirely omitted 'All' the literature in ET in hydrology'. With all due respect, that's incorrect and 'in contradiction' with the Referee's own words (see the underlined first part of the second sentence of the Referee's comment above). In contrast to the Referee's incorrect statement, we had many key references on the evapotranspiration in the Introduction and also in pages 16 through 18 (under the 'Future Importance' section) of the original manuscript.

Please note that this is not a review manuscript on evapotranspiration, yet we will be citing over 80 references in the revised manuscript, including the new references in blue-colored new paragraphs in our Response to Comment #8. Besides, we refereed to McMahon et al. (2013)'s paper in our manuscript for more comprehensive discussion on different evapotranspiration measures. Our manuscript rather aims at introducing and briefly describing different evapotranspiration measures, prediction using a novel probabilistic ML framework and standardized climate data, and assess the accuracy, robustness, and prediction uncertainties using the PME-computed $ET_o$, upscaled $E_{sw}$ data, and measured $ET_a$ data from the EC tower.

So, the main objectives (as already stated in the Abstract, Introduction, main body, and conclusions sections of the manuscript) is development of a new data-driven, probabilistic **ML method** to predict $ET_o$, $E_{sw}$, and $ET_a$ by avoiding computationally involved net solar radiation calculations, overcoming uncertainties in pan evaporation and pan coefficients, and offsetting high capital and maintenance costs in $ET_a$ measured from an EC tower (already described in the Introduction, main body, and Conclusion sections of the manuscript).

Having said this, we are still open to any references (not included in the original manuscript and the blue-marked new sections) the Referee may suggest.

*9.1 'Although the authors cited some ML for ET papers, they were discussed very surficially '*

We respectfully disagree with this comment. The purpose of citations for ML-based ET prediction (in line #45−57 on page 2) to show that various ML models/methods have been developed and used for ET prediction, **but without accounting for inherit prediction uncertainties**. This issue was brought up recently by Tang et al. (2018), as we stated in the Introduction section, and we confronted this challenge in our manuscript for the first time in predicting $ETo$, $ET_a$, and $E_{sw}$ prediction. This laid the foundation for motivation (iv), as further elaborated in our response 1.1 above and Objective 2 in the Introduction section of the original manuscript.

Because this is not a review paper, we needed to extract the **key points** from the references to lay out the main foundation for the main objectives and motivation for this work, and our review of the ML-based ET papers accomplished this; therefore, we do not consider the discussion 'surficial', as the Referee suggested.
* * *
**Comment 10.** Then the authors spent 8 figures presenting stuff that has no modeling components. ?

**Response:**

'8 figures with no modeling components' in the comment above is an inaccurate statement.

We had 12 figures in the original manuscript, excluding the figures in Appendix). Among them, Fig. 5 (modeled by the PME), Fig. 6 (modeled by MF), Fig. 7 (modeled by the PME), Fig. 8 (modeled by the PME), Fig. 9 (modeled by hybrid XGBoost-NGBoost), Fig. 10 (modeled by hybrid XGBoost-NGBoost and Shapley Values method) and Fig. 12 (modeled by hybrid XGBoost-NGBoost and Shapley Values method).

Therefore, the number of figures with 'no direct modeling component' is 5 out of 12 (40% of the total figures). The remaining figures display the location map, input data for the models, and correlation maps to interpret data, which are still related to the model setups.

Having said this, we did not see any requirement in the manuscript preparation guideline for authors about the required number of figures that need to allocated to the 'modeling component'. We believe that the figures in our manuscript enhances its transparency by introducing the readers to the data that we used for modeling

and the modeling results.
* * *
**Comment 11.** It was very hard to understand what were the inputs and what were the outputs from Methods. This was instead discussed in the Results section.

**Response:**

Input and output data were described in the original manuscript far before the 'Results' section. For example, as described in line $\#198-199$ of the manuscript (section 2.1 on page 7), $T_a$, $P$, $RH$, $u_2$, and $R_s$ are the **input** data for the PME (and hence, for the probabilistic ML modeling used for $ET_o$ prediction). As described in line $\#215$, $T_a$ (input data) was replaced by $T_{SW}$ (another input) in $E_{sw}$ calculations. In the original manuscript, the section entitle 'Data Availability' (along with the Appendix) is dedicated for the input data. Moreover, Shapley plots in Fig. 10 list all the input data for the probabilistic ML modeling along the y-axis, as the method quantifies the importance and interaction of each input feature on $ET_o$, $E_{sw}$, and $E_a$.

However, we see the concern raised by the Referee. Therefore, to enhance the clarity about the inputs and outputs, we will replace Fig. 1 in the original manuscript with the new figure in Fig. 2 below.

[Figure]

Figure 2: Conceptual representation of the hybrid NGBoost-XGBoost model for $ET_o$, $E_{sw}$, and $ET_a$ prediction.
* * *
**Comment 11.** The authors used 'unprecedented' and 'for the first time' very casually. There are things we do and things we don't do and there must be a clear reason why we do them.

**Response:**

Based on our current literature review, we use the terms 'unprecedented' and 'for the first time' very meticulously (not causally at all) throughout the manuscript. However, if the Referee has any compelling and convincing references and 'reason' (not included in the Referee's comment above) that may invalidate our use of these terms in the sentences below, we kindly ask the Referee to share these references and information with us for our review.

*... Unprecedentedly, we demonstrate that a newly developed probabilistic machine learning (ML) model, using a hybridized NGBoost-XGBoost framework, can accurately predict the daily $ET_o$, $E_{sw}$, $ET_a$ from local climate data. The probabilistic approach exhibits great potential in overcoming data uncertainties, in which 99% of the ETo, 90% of the $E_{sw}$, and 91% of the $ET_a$ test data at three watersheds were within the model's*

*95% prediction interval....*

**Question to Referee:** If the Referee thinks that we use the term 'unprecedented' causally in the sentence above, we would like to know which journal article has already reported development and use of a probabilistic ML model to predict $ET_o$, $E_{sw}$, $ET_a$ from local climate data within $\geq 90\%$ within the model's 95% prediction interval?

also

  *... Finally, we demonstrate, for the first time, a coalition game theory approach to identify the order of importance, dependencies & interactions of climatic variables on the ML-based $ET_o$, $E_{sw}$, $ET_a$ predictions....*

**Question to Referee:** If the Referee thinks that we use the term 'for the first time' causally in the sentence above, we would like to know which journal article reports the use of the coalition game theory approach to determine the order of importance, dependencies & interactions of climatic variables on the ML-based $ET_o$, $E_{sw}$, and $ET_a$ predictions?